# Altered G1 signaling order and commitment point in cells proliferating without CDK4/6 activity

Chad Liu [1], Yumi Konagaya [1,2,3], Mingyu Chung[1], Leighton H. Daigh [1], Yilin Fan [1], Hee Won Yang[1,5], Kenta Terai[2], Michiyuki Matsuda [2,4] & Tobias Meyer [1,3 ✉]

Cell-cycle entry relies on an orderly progression of signaling events. To start, cells first activate the kinase cyclin D-CDK4/6, which leads to eventual inactivation of the retinoblastoma protein Rb. Hours later, cells inactivate APC/C$^{CDH1}$ and cross the final commitment point. However, many cells with genetically deleted cyclin Ds, which activate and confer specificity to CDK4/6, can compensate and proliferate. Despite its importance in cancer, how this entry mechanism operates remains poorly characterized, and whether cells use this path under normal conditions remains unknown. Here, using single-cell microscopy, we demonstrate that cells with acutely inhibited CDK4/6 enter the cell cycle with a slowed and fluctuating cyclin E-CDK2 activity increase. Surprisingly, with low CDK4/6 activity, the order of APC/C$^{CDH1}$ and Rb inactivation is reversed in both cell lines and wild-type mice. Finally, we show that as a consequence of this signaling inversion, Rb inactivation replaces APC/C$^{CDH1}$ inactivation as the point of no return. Together, we elucidate the molecular steps that enable cell-cycle entry without CDK4/6 activity. Our findings not only have implications in cancer resistance, but also reveal temporal plasticity underlying the G1 regulatory circuit.

[1] Department of Chemical and Systems Biology, Stanford Medicine, Stanford, CA 94305, United States. [2] Laboratory of Bioimaging and Cell Signaling, Graduate School of Biostudies, Kyoto University, Kyoto 606-8501, Japan. [3] Department of Cell and Developmental Biology, Weill Cornell Medicine, 1300 York Ave, New York, NY 10065, USA. [4] Department of Pathology and Biology of Diseases, Kyoto University, Kyoto, Japan. [5]Present address: Department of Pathology and Cell Biology, Columbia University Medical Center, 630 West 168th Street, New York, NY 10032, USA. ✉email: tom4003@med.cornell.edu

To exit quiescence and start the cell cycle, non-embryonic cells first activate G1 cyclin dependent kinases (CDKs), then suppress Rb function, and finally, inactivate the E3 ubiquitin ligase APC/C[CDH1] to trigger irreversible commitment[1,2]. More specifically, cells initiate the cell cycle by upregulating cyclin D to activate CDK4 and CDK6 (hereafter CDK4/6), resulting in Rb phosphorylation and eventual Rb inactivation[3–7]. In turn, Rb inactivation leads to the upregulation of critical E2F targets such as the CDK2 activator cyclin E, APC/C[CDH1] inhibitor Emi1, and various factors needed to replicate DNA and prevent DNA damage[8–10]. Cyclin E and Emi1 accumulation results in APC/C[CDH1] inactivation a few hours later, which not only prepares cells metabolically for S phase, but allows for the buildup of cyclin A to promote DNA replication[11,12]. Furthermore, once APC/C[CDH1] inactivation is initiated, cell-cycle entry becomes irreversible in respect to various types of stress, reflecting an underlying G1 commitment point (Fig. 1a, top)[13–15].

When D-class cyclins are genetically ablated, however, many non-transformed cells can still start the cell cycle by directly activating cyclin E-CDK2[16]. Notably, deletion of both D and E-class cyclins blocks cell-cycle progression in MEFs and most cell lineages in a developing embryo[17]. These results suggest that in many non-transformed cells, E-class cyclins can compensate if D-class cyclins are missing. While the canonical cyclin D-CDK4/6-initiated entry mechanism has been extensively characterized in a variety of contexts, the alternate cyclin E-CDK2-initiated entry mechanism has mainly been characterized using knockout models and in cancer, where cells can bypass CDK4/6 inhibition via c-Myc upregulation of cyclin E-CDK2 activity, amplification of cyclin E, or downregulation of CDK2 inhibitors[18–25]. However, since the cyclin E-CDK2-initiated entry mechanism has only been described in cancer and in scenarios where cyclin Ds are deleted at germline, it remains unknown whether wild-type cells can make use of this mechanism. Furthermore, it is generally assumed that cyclin E-CDK2-initiated cell-cycle entry requires a set order of events where cyclin E-CDK2 activation is followed by Rb inactivation, which is then followed by APC/C[CDH1] inactivation and irreversible commitment. This hypothesis of a rigid order underlying G1 progression has also not been experimentally tested (Fig. 1a, bottom).

Here, applying live and fixed single-cell microscopy, we show that cells with acutely inhibited CDK4/6 activity can still proliferate but with cyclin E-CDK2 being initially activated non-persistently and without Rb hyperphosphorylation. Rb is eventually inactivated prior to DNA replication, but surprisingly, the order of Rb and APC/C[CDH1] inactivation is reversed both in non-transformed cell lines as well as in the small intestinal crypts of wild-type mice. Thus, our study argues that cells can enter the cell cycle without CDK4/6 activity also under normal conditions and further argues against a rigid order of signaling events in G1. Finally, we show that this signaling inversion leads to a point of no return that is marked by Rb inactivation instead of APC/C[CDH1] inactivation. Thus, to start the cell cycle, cells can first activate CDKs via upregulation of D or E-class cyclins, then either inactivate Rb before APC/C[CDH1] or inactivate APC/C[CDH1] before Rb in an interchangeable manner, and finally, commit to the cell cycle only after both Rb and APC/C[CDH1] are inactivated.

## Results

### Acute CDK4/6 inhibition reveals a delayed and less efficient cell-cycle entry mechanism.

To monitor cell-cycle entry at the single-cell level, we stably transduced non-transformed MCF-10A epithelial cells with a previously characterized APC/C[CDH1] activity reporter that is degraded during G0 and G1 and accumulated linearly during late G1, S, and G2 phase (Fig. 1b, left)[14,26]. We deprived cells of growth factors for 48 h, and then added back mitogen in the presence or absence of the CDK4/6 inhibitor Palbociclib[27,28] (hereafter just CDK4/6 inhibitor) along with EdU, a nucleoside analog that is incorporated by cells into replicated DNA during S phase. Markedly, we found that a fraction of CDK4/6 inhibitor-treated cells still inactivated APC/C[CDH1] and incorporated EdU (Fig. 1b, right; Fig. 1c), arguing that post-embryonic cells can enter S phase without cyclin D-CDK4/6 activity even when cyclin Ds are not deleted at germline (the CDK4/6 inhibitor-treated cells eventually undergo mitosis). We reason that if there is residual CDK4/6 activity that is driving cells into S phase, refreshing the drug more frequently or titrating up the drug should decrease the percentage of S-phase cells. However, entry into S phase was independent of the drug refreshing rate and was also not further inhibited by a threefold increase in inhibitor concentration (Fig. 1d, we note that Palbociclib at 1 μM already inhibits phosphorylation of Rb, CDK4/6's best characterized substrate[7,27]). Given that cells can still enter the cell cycle at 3 μM Palbociclib, we deduce that there is minimal functional contribution of CDK4/6 activity to S-phase entry under these conditions. Cell-cycle entry in CDK4/6-inhibited cells was notably delayed and less synchronized compared to cells with intact CDK4/6 activity, demonstrating that these cells have an extended G1 phase that greatly varies between cells. When tracking cells for three days, we found that a majority of them eventually inactivated APC/C[CDH1] (Fig. 1e). Since MEFs genetically depleted of cyclin Ds also exhibit less efficient entry into S phase as well as a longer G1[16], we reasoned that acute chemical inhibition of CDK4/6 phenocopies the ablation of cyclin Ds, suggesting that this CDK4/6 activity-independent cell-cycle entry mechanism is more general and not only the result of a long-term compensation mechanism induced by germline cyclin D loss.

We next broadened the analysis to asynchronously cycling cells, where most newborn daughter cells do not enter quiescence and start preparing and committing to the cell cycle in mother cells[1,29–31]. By in silico aligning cells at the time of birth, we found the same extended G1 phase also in CDK4/6-inhibited cycling cells (Fig. 1f; CDF: Supplementary Fig. 1a). Furthermore, acute CDK4/6 inhibitor treatment reduced the percentage of cells in S/G2 in a variety of normal and cancer cells, suggesting that G1 extension and less efficient cell-cycle entry are general phenomena in CDK4/6-inhibited cells (Fig. 1g). We note that cells need intact Rb inhibitory function to be responsive to CDK4/6 inhibition, explaining the lack of effect of CDK4/6 inhibition in HeLa cells that have genetically inactivated Rb (Fig. 1g)[24]. Finally, even though sister cells mostly had the same fate after birth (Supplementary Fig. 1b), live-cell analysis showed that subsequent generations of CDK4/6-inhibited cells were not significantly more likely to re-enter the cell cycle compared to the first generation (experiment setup: Supplementary Fig. 1c; results: Fig. 1h). This suggests that subsequent generations are not clonal subpopulations of MCF-10A cells that gained heritable resistance to CDK4/6 inhibition (like HeLa cells did).

### Cells without CDK4/6 activity exhibit slower and fluctuating increases in cyclin E-CDK2 activity.

Because cells without CDK4/6 activity rely on cyclin E-CDK2 for cell-cycle entry, we stably expressed in the same cells a previously characterized live-cell cyclin E/A-CDK activity reporter based on a DHB peptide (Fig. 2a)[32–34]. We note that the reporter detects primarily cyclin E-CDK2 activity in G1 since there is minimal cyclin A protein or CDK1 activity in G1[14,33,35,36]. This reporter measures relative kinase activity as the ratio of the cytoplasmic over nuclear reporter concentrations. The activity ratio increases from approximately 0.5 at basal, to 1 at the G1/S transition, and to 1.5

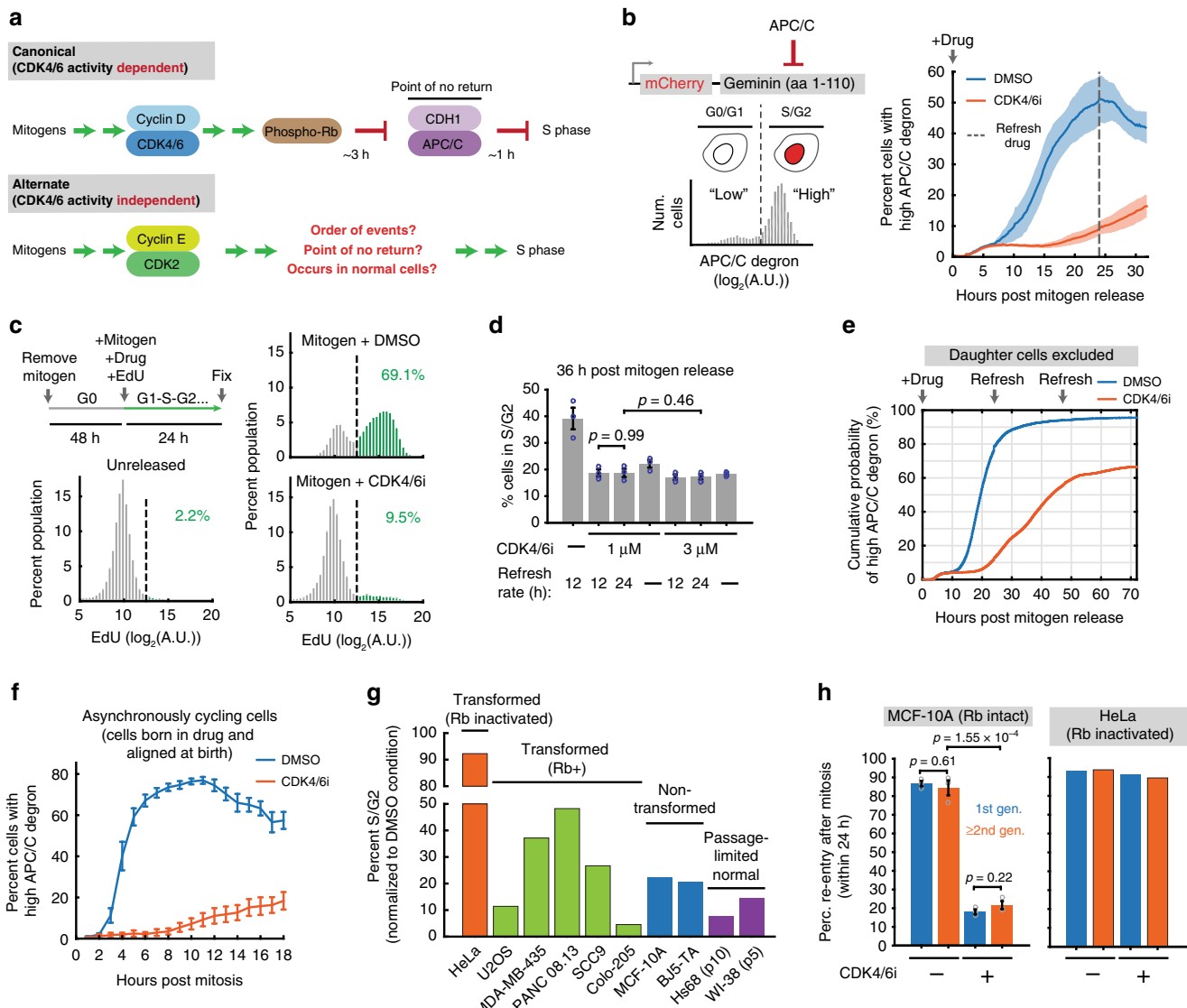

**Fig. 1 Acute CDK4/6 inhibition reveals a delayed and less efficient cell-cycle entry mechanism. a** Two mechanisms of starting the cell cycle. **b** Left: Degron-based APC/C activity reporter. Right: Mitogen-starved MCF-10A cells expressing the APC/C activity reporter were imaged at the time of mitogen and 1 μM Palbociclib (CDK4/6 inhibitor) addition. Time points taken every 12 min. CDK4/6 inhibitor refreshed at 24 h. Mean ± SEM from five biological replicates. Vertical gray dashed line: time of refreshing drug. **c** MCF-10A cells mitogen released in the presence of EdU with or without CDK4/6 inhibitor. Cells were fixed after 24 h (26838, 36549, 29371 cells for unreleased, DMSO, and CDK4/6 inhibitor; 1 of $n = 2$ biological replicates). **d** Cells were mitogen released with or without indicated CDK4/6 inhibitor concentrations and the drug was refreshed every 12 h, 24 h, or not at all. Cells fixed at 36 h after mitogen release. S/G2 cells determined via EdU incorporation and DNA content. Mean ± SEM from three biological replicates; $p$ values calculated using two-sided, two-sample $t$ tests. **e** CDF of mitogen-released cells inactivating APC/C$^{CDH1}$. Only cells present at the time of mitogen release are tracked. $n = 7521$ and 9563 cells for DMSO and CDK4/6i, respectively. **f** Percent cells with inactivated APC/C in cells born into DMSO or CDK4/6 inhibitor. Cells aligned at birth. Mean ± SEM from five biological replicates. **g** Different cell lines treated with 1 μM CDK4/6 inhibitor for 48 h and refreshed at 24 h. Percent S/G2 defined as >2$n$ DNA (determined by Hoechst) and high endogenous geminin (an APC/C$^{CDH1}$ substrate). Percentages normalized to the DMSO-treated condition. **h** Generation comparison of cell-cycle re-entry percentages in MCF-10A (Rb intact) and HeLa (Rb inactivated) cells (MCF10A: mean ± SEM from three biological replicates; $p$ values calculated using two-sided, two-sample $t$ tests).

when cells enter mitosis. Starting live-cell imaging at the time of mitogen release to determine the cyclin E/A-CDK activity baseline, we found that a portion of CDK4/6 inhibitor-treated cells indeed upregulated cyclin E-CDK2 activity after variable delays (Fig. 2b, blue cells, which is defined as >0.7 on the last time point; 0.7 threshold marked with horizontal gray dashed line and determined based on time 0, when no CDK activity is present). As expected, activation of cyclin E-CDK2 was delayed in CDK4/6-inhibited cells (Supplementary Fig. 2a), but surprisingly, the kinetics were distinct from that of control cells, where cyclin E/A-

CDK activity in most cells increased in a persistent and linear fashion (Fig. 2b, left). An analysis of thousands of individual cells showed that CDK4/6-inhibited cells built up cyclin E/A-CDK activity slower (Fig. 2b, blue cells) and some cells had fluctuating activities (green cells, defined as <0.7 at 24 h, but >0.7 at least once between 8 and 24 h). The increasing trend of fluctuating cells in CDK4/6-inhibited cells is not due to threshold selection (Supplementary Fig. 2b). Further control experiments with a DHB mutant that cannot translocate showed that most fluctuations were not measurement noise, and that the fluctuating cells

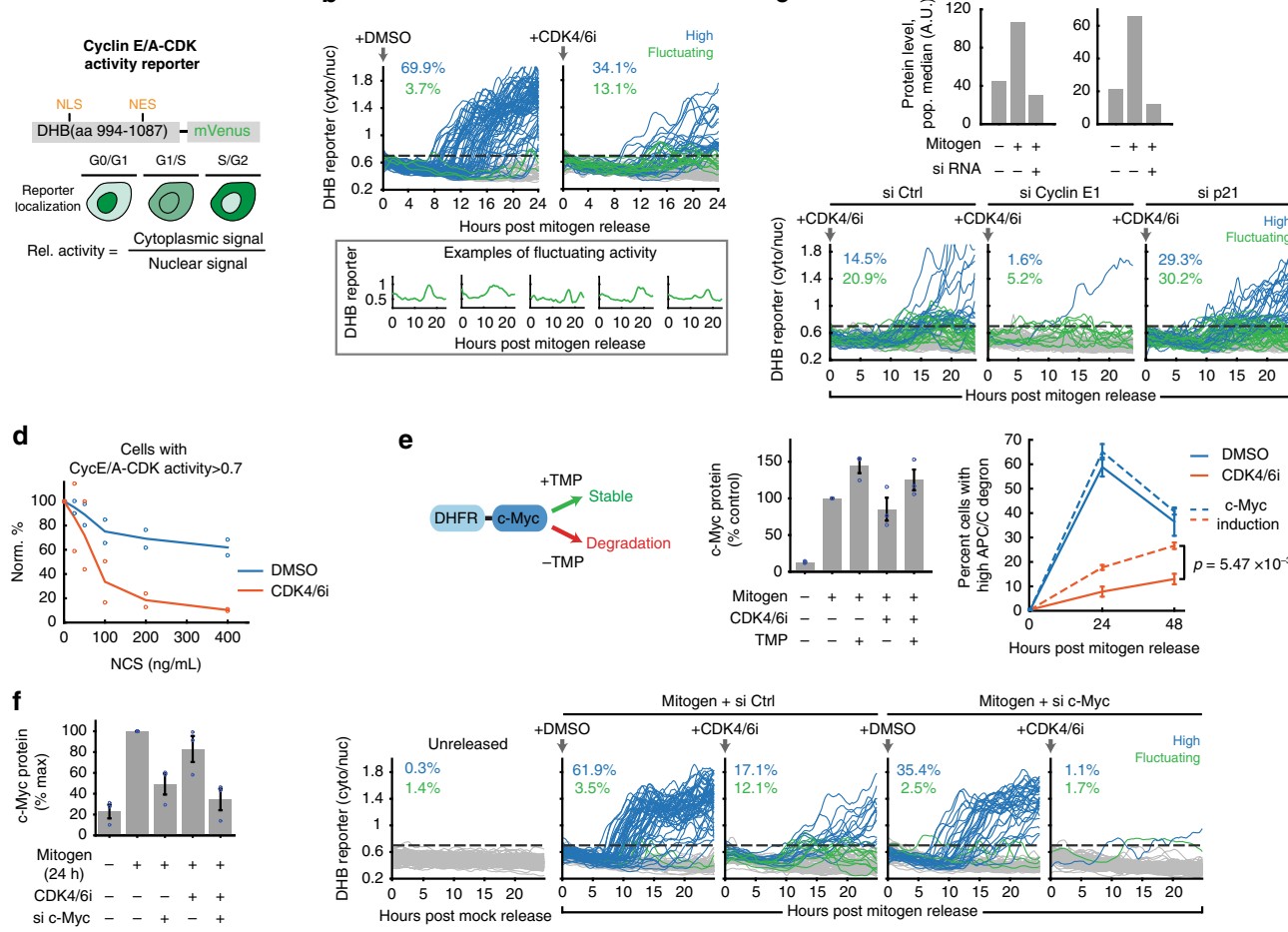

**Fig. 2 Cells without CDK4/6 activity exhibit slower and fluctuating increases in cyclin E-CDK2 activity. a** DHB-based cyclin E/A-CDK activity reporter. **b** Single-cell traces of cyclin E/A-CDK activity after mitogen release with or without 1 μM CDK4/6 inhibitor (100 cells plotted; 2782 and 8548 cells total for DMSO and CDK4/6 inhibitor; 1 of $n = 2$ biological replicates). Activity baseline at time 0 is determined to be at most 0.7 (dashed horizontal line). Blue: >0.7 at last time point. Green: <0.7 at last time point, exceeded 0.7 at some point between 8 and 24 h. Gray: rest of the cells. Bottom: Examples of fluctuating cyclin E/A-CDK activity. **c** Top: immunofluorescence validation of cyclin E1 and p21 knockdown. Bottom: effect of knockdown on cyclin E/A-CDK activity in cells treated with CDK4/6 inhibitor. Traces classified the same way as (**b**) (100 cells plotted; 2958, 3707, 3113 cells for si Ctrl, si Cyclin E1, and si p21; 1 of $n = 2$ biological replicates). **d** Cells were treated with neocarzinostatin (NCS) for 15 min, and then mitogen released for 24 h. Percent cells with high cyclin E/A-CDK activity (defined as reporter ratio >0.7) are displayed. Percentages normalized to 0 ng/mL condition. Mean of $n = 2$ biological replicates. **e** Cells were stably infected with DHFR-Myc that is unstable until TMP molecule addition. c-Myc protein levels were induced to ~50% above control after adding 10 μM TMP at the time of mitogen release. Cells fixed at 24 and 48 h after mitogen release. c-Myc protein level measured via immunofluorescence. Drugs refreshed every 24 h. Mean ± SEM from three biological replicates; $p$ value calculated over the 48 h CDK4/6i condition using a two-sided, two-sample $t$ test. **f** Left: immunofluorescence of c-Myc knockdown validation (mean ± SEM from three biological replicates). Right: cyclin E/A-CDK activity after mitogen release in MCF-10A (100 cells plotted, 579, 3446, 3904, 3935, 3767 total cells for unreleased, si Ctrl+DMSO, si Ctrl+CDK4/6 inhibitor, si c-Myc+DMSO, and si c-Myc+CDK4/6 inhibitor, respectively), 1 of $n = 3$ biological replicates.

had not started S phase (Supplementary Fig. 2c, d). In addition, treatment with a saturating dose of a CDK1/2 inhibitor showed identical sensor kinetics in cells proliferating with and without CDK4/6 activity, demonstrating that the sensor fluctuation is due to kinase activity and not change in phosphatase activity (Supplementary Fig. 2e).

The increase in cyclin E/A-CDK activity was regulated by cyclin E and the CDK2 inhibitor p21 (western blot knockdown validation: Supplementary Fig. 2f; immunofluorescence knockdown validation and live-cell experiment: Fig. 2c—we note that the transfection protocol can lower the overall percentages of cells that enter the cell cycle). These results suggest that the CDK activity fluctuations and increases result from competition between dynamically changing cyclin E or p21 levels. Because stress is known to upregulate p21 and downregulate CDK2 activity[14,37], we hypothesize that cells with low CDK4/6 activity

are more sensitive to stress given its slow and fluctuating CDK2 activity. Indeed, a 15-min pulse of neocarzinostatin (can cause DNA breaks in quiescence[38]) prior to mitogen release confirmed that CDK2 activity in cells without CDK4/6 activity is more strongly affected compared to cells with CDK4/6 activity (Fig. 2d).

Previous reports showed that c-Myc overexpression can promote S-phase entry in cells where CDK4/6 activity was inhibited by p16 expression[18]. We confirmed that a 50% increase of c-Myc protein levels was sufficient to significantly increase the fraction of CDK4/6-inhibited cells entering the cell cycle (Fig. 2e). We also note that c-Myc protein level is not significantly affected by CDK4/6 inhibition (Fig. 2e, protein measurement in no TMP condition; Supplementary Fig. 3a). Furthermore, c-Myc knockdown in the presence of CDK4/6 inhibitor reduced cyclin E transcription and abolished cyclin E-CDK2 activity and

APC/C[CDH1] inactivation (western knockdown validation: Supplementary Fig. 3b; cyclin E transcription: Supplementary Fig. 3c; immunofluorescence knockdown validation and live-cell experiment: Fig. 2f; APC/C[CDH1] inactivation: Supplementary Fig. 3d). We conclude that c-Myc and cyclin D-CDK4/6 activities are driving forces that can promote cell-cycle entry through different initiation mechanisms—one that results in a slower and often fluctuating initial increase in CDK2 activity, and one that results in a faster and more persistent increase in CDK2 activity. In addition, our experiments revealed that a key role of CDK4/6 is to confer persistence to cyclin E-CDK2 activation in G1.

**Cyclin E-CDK2 is activated without Rb hyperphosphorylation in cells lacking CDK4/6 activity**. To understand the role and timing of Rb inactivation in G1 cells with inhibited CDK4/6 activity, we examined the phosphorylation status of Rb. When phosphorylated on most or all of its accessible sites ("hyperphosphorylation"), Rb releases E2F transcription factors that upregulate a large set of cell-cycle progression genes (Fig. 3a)[4,5,8,9]. We first restricted our live-cell analysis to G1 cells at early stages of CDK2 activation by selecting cells that had not yet inactivated APC/C[CDH1] and had just activated cyclin E-CDK2. We then fixed cells and measured the phosphorylation status of serine 807 and serine 811 (S807/S811) on Rb by normalizing the immunofluorescent phosphorylation signal against that of total Rb in each cell (Fig. 3b). Analysis of DMSO-treated control cells that had just activated cyclin E-CDK2 showed that these cells already have S807/S811 phosphorylated at levels nearly identical to that of S/G2 cells, when Rb is hyperphosphorylated (Fig. 3c, top and middle)[5]. This is consistent with recent work from our laboratory demonstrating that this phospho-Rb (S807/S811) immunofluorescence signal can be used as a marker for Rb hyperphosphorylation (but cannot distinguish between unphosphorylated and mono-phosphorylated Rb, two Rb forms that still bind E2Fs)[7]. However, unexpectedly, in CDK4/6-inhibited cells, we observed a much lower signal of phospho-S807/S811 even though the measured cyclin E/A-CDK activity level was the same (Fig. 3c, bottom). Control experiments showed the same result of suppressed Rb phosphorylation in CDK4/6 inhibitor-treated BJ-5ta, a non-transformed human foreskin fibroblast (Fig. 3d), as well as when we used two alternative CDK4/6 inhibitors with different pharmacological properties (Supplementary Fig. 4). Since Rb hyperphosphorylation and inactivation requires most or all sites to be phosphorylated[5], these results indicate that early cyclin E-CDK2 activation in CDK4/6-inhibited cells does not require Rb hyperphosphorylation. CDK2 activation is therefore the result of direct c-Myc, rather than E2F, mediated upregulation of cyclin E.

Markedly, when we plotted the percentages of cells with high phospho-Rb (S807/S811) signal as a function of cyclin E/A-CDK activity, we found that CDK4/6-inhibited cells did eventually reach the high phospho-Rb state, but only at much higher cyclin E/A-CDK activity levels (Fig. 3e). Thus, Rb inactivation may occur much later in cells without CDK4/6 activity. This delay is likely due to CDK4/6's ability to contribute to or initially mediate Rb hyperphosphorylation[3–7], and also argues that much higher cyclin E/A-CDK activity is required in CDK4/6-inhibited cells to reach maximal Rb phosphorylation. When we gated for CDK4/6-inhibited cells that have already entered S phase and measured phospho-Rb signal after treatment with CDK1/2 and CDK1 inhibitors, the high phospho-Rb signal disappeared after incubation with CDK1/2 inhibitor but not CDK1 inhibitor (Fig. 3f). Together, we conclude that in CDK4/6-inhibited cells, Rb is phosphorylated only in late G1 at much higher levels of CDK2 activity compared to cells that activate CDK4/6.

**The canonical signaling order of Rb inactivation before APC/C[CDH1] inactivation is reversed in cells proliferating without CDK4/6 activity**. Due to the need for higher CDK2 activity for Rb phosphorylation in cells without CDK4/6 activity (Fig. 3e), we hypothesized that, in the presence of CDK4/6 inhibitor, full Rb phosphorylation and inactivation may occur only after APC/C[CDH1] inactivation. This is possible since APC/C[CDH1] inactivation can be mediated by cyclin E-CDK2 phosphorylation of Cdh1 even in the absence of APC/C[CDH1] inhibitor Emi1[14,15]. Indeed, APC/C[CDH1] inactivates at approximately the same level of cyclin E/A-CDK activity with or without CDK4/6 activity (Supplementary Fig. 5a). To directly determine the relative order of events during G1, we measured high phospho-Rb (S807/S811) signal in cells that had just inactivated APC/C[CDH1] using the increase in the APC/C degron signal as a proxy for the time after APC/C[CDH1] inactivation (Fig. 4a, left; we note that different time points were chosen to maximize number of cells since APC/C[CDH1] inactivation is delayed in CDK4/6-inhibited cells). Such a conversion from concentration to time is possible since the concentration of the APC/C degron reporter increases in a linear fashion after APC/C[CDH1] inactivation. Consistent with our previous results, nearly all of the DMSO-treated cells that just inactivated APC/C[CDH1] already had high phospho-Rb(S807/S811) signal (Fig. 4a, DMSO histograms). Strikingly, in CDK4/6-inhibited cells, the high Rb phosphorylation signal peak in the histogram only started to appear after a delay following the buildup of APC/C degron (Fig. 4a, CDK4/6i histograms). The same delay also occurs in asynchronously cycling cells (Supplementary Fig. 5b). Thus, the order of Rb inactivation and APC/C[CDH1] inactivation is reversed in CDK4/6-inhibited cells with Rb inactivation occurring only after APC/C[CDH1] inactivation.

To directly measure whether E2F targets are also only activated after APC/C[CDH1] inactivation in CDK4/6-inhibited cells, we measured the transcripts of E2F1, a well-characterized E2F target that is suppressed by Rb[9]. Using single-cell mRNA fluorescence in situ hybridization (FISH) analysis, we confirmed that E2F1 transcription was suppressed by CDK4/6 inhibition in cells with activated cyclin E-CDK2 and active APC/C[CDH1] (Fig. 4b, left; Supplementary Fig. 5c, left). Furthermore, E2F1 transcription increased in CDK4/6-inhibited cells only after both cyclin E-CDK2 activation and APC/C[CDH1] inactivation (Fig. 4b, right; Supplementary Fig. 5c, right).

To further validate that Rb is inactivated only after APC/C[CDH1] inactivation, we made use of previous findings that inactivated Rb dissociates from the chromatin[39]. We thus measured pre-extracted/chromatin-bound Rb in single cells[40]. Control experiments confirmed that in cells with CDK4/6 activity, Rb dissociates from chromatin (low chromatin-bound Rb signal) before APC/C[CDH1] inactivation and remains dissociated after APC/C[CDH1] is turned off (Fig. 4c, DMSO condition, green and blue arrows; BJ-5ta cells: Supplementary Fig. 6a). However, in CDK4/6-inhibited cells, Rb only dissociated from chromatin after a delay and after significant APC/C degron buildup (Fig. 4c, CDK4/6 inhibitor condition, orange and green arrows; quantification for BJ-5ta and MCF10A cells: Supplementary Fig. 6b, c). Together, our results of delayed Rb phosphorylation, Rb chromatin dissociation, and E2F1 activation demonstrate that CDK4/6-inhibited cells have inverted G1 signaling, with cyclin E-CDK2 being activated first, and APC/C[CDH1] then being inactivated before Rb inactivation—while Rb is inactivated before APC/C[CDH1] in cells with active CDK4/6.

To test whether earlier APC/C[CDH1] inactivation in CDK4/6-inhibited cells influences the inactivation of Rb, we knocked down Cdh1 to accelerate APC/C[CDH1] inactivation, and found that the percentage of cells with Rb inactivation increased (western knockdown validation: Supplementary Fig. 7a; phospho-Rb

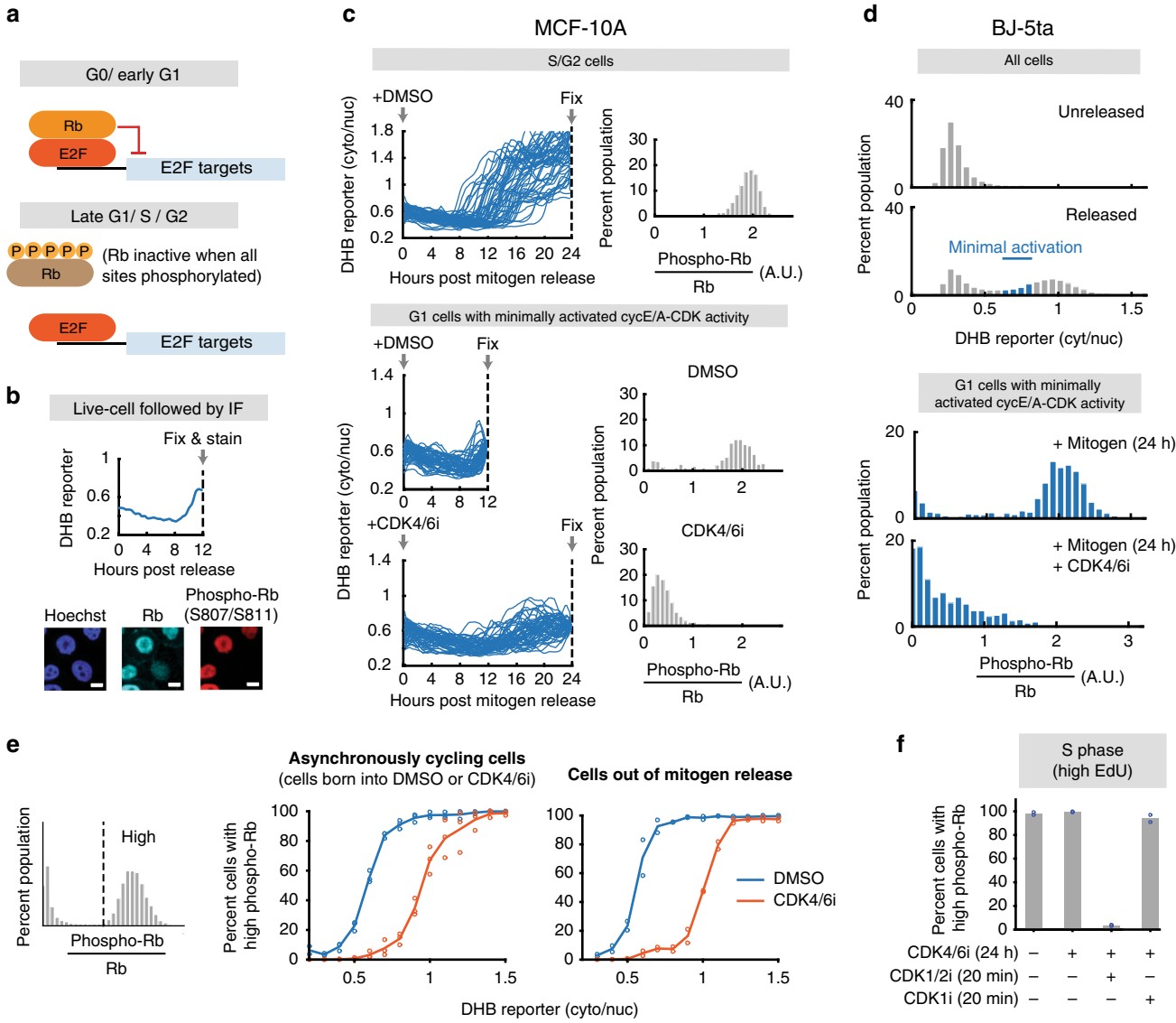

**Fig. 3 Cyclin E-CDK2 is activated without Rb hyperphosphorylation in cells lacking CDK4/6 activity. a** E2F is active when Rb is hyperphosphorylated. **b** Phospho-Rb analysis after live-cell tracking of cyclin E/A-CDK activity. Scale bar = 10 μm. **c** Phospho-Rb(S807/S811) analysis of cells that have inactivated APC/C$^{CDH1}$ (top; 50 cells plotted, 2315 cells total), and cells that have initiated cyclin E/A-CDK activity and have not inactivated APC/C$^{CDH1}$ (bottom; 50 cells plotted, 364, 1023 cells for DMSO and CDK4/6 inhibitor). One of $n = 3$ biological replicates. **d** BJ-5ta cells expressing the cyclin E/A-CDK activity reporter and APC/C degron reporter were mitogen released with or without 1 μM CDK4/6 inhibitor, fixed after 24 h, and stained for Rb and phospho-Rb(S807/S811). G1 cells with recently activated cyclin E/A-CDK activity were analyzed. $n = 703$ and 383 cells for DMSO and CDK4/6i condition. **e** Cells were grouped into high and low phospho-Rb(S807/S811) signal populations. The percentages of cells with high signal were then determined for each bin of the cyclin E/A-CDK activity. Middle: Cells born into DMSO or CDK4/6 inhibitor. Mean of $n = 3$ biological replicates. Right: Cells mitogen released with DMSO or CDK4/6 inhibitor. Mean of $n = 2$ biological replicates. **f** Asynchronously cycling cells treated with or without CDK4/6 inhibitor for 24 h. Cells were then incubated with 10 μM EdU for 15 min, and then treated with DMSO, CDK1/2i (3 μM), or CDK1i (10 μM) for 20 min. Cells were then fixed and high EdU signal cells were examined. Mean of $n = 2$ biological replicates.

measurements: Supplementary Fig. 7b). This suggests that APC/C$^{CDH1}$ inactivation may facilitate Rb inactivation in CDK4/6-inhibited cells, perhaps through increasing CDK2 activity (Supplementary Fig. 7b, right most panels). This observation is also consistent with previous report that Cdh1 knockdown confers partial cancer resistance to CDK4/6 inhibition[41]. Thus, in cells lacking CDK4/6 activity, not only is the order of APC/C$^{CDH1}$ and Rb inactivation reversed, the role of Rb inactivation promoting APC/C$^{CDH1}$ inactivation may be reversed as well, where instead, APC/C$^{CDH1}$ inactivation promotes Rb inactivation.

**A fraction of small intestinal crypt cells have the unphosphorylated Rb and inactive APC/C$^{CDH1}$ characteristic of cells entering the cell cycle without CDK4/6 activity.** We next examined whether normal cells in vivo can start the cell cycle using the CDK2-initiated path by analyzing mouse small intestinal crypts, a proliferative region where cells are continuously replenished (Fig. 5a, b)[42]. Based on our cell line data, we know that cells in mid G1 are (1) CDK4/6-initiated when they have high phosphorylated Rb signal and low APC/C degron expression and (2) CDK2-initiated when they have high APC/C degron expression and low Rb phosphorylation signal. We note that for

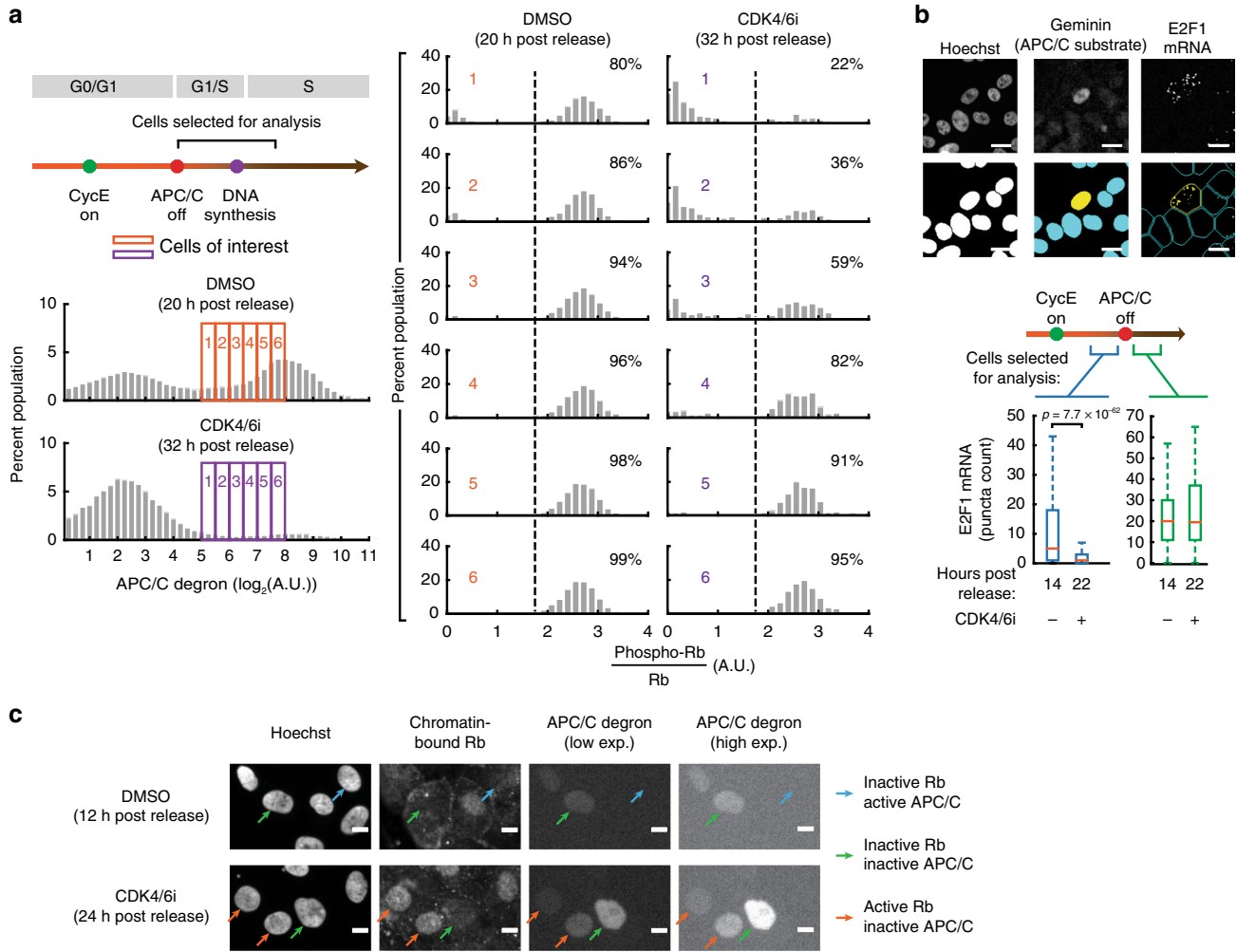

**Fig. 4 The canonical signaling order of Rb inactivation before APC/C^CDH1 inactivation is reversed in cells proliferating without CDK4/6 activity. a** Cells mitogen released with DMSO or 1 μM CDK4/6 inhibitor and have recently inactivated APC/C^CDH1 were analyzed for phospho-Rb(S807/S811) signal (>800 cells per histogram for DMSO > 200 cells per histogram for CDK4/6 inhibitor; 1 of $n = 3$ biological replicates). **b** Top: Single-cell mRNA quantification (scale bar = 10 μm). Bottom: E2F1 mRNA in cells released with or without CDK4/6 inhibitor. High cyclin E/A-CDK activity defined as ratio of signal >0.7. $p$ values calculated using two-sided, two-sample $t$ tests (from left to right: $n = 784, 1484, 638$, and 98 cells). Box center indicates median, box edges denote quartiles, and whiskers extend to 1.5 times the interquartile range away from the box edges. One of $n = 3$ biological replicates. **c** Mitogen-released cells were assayed for chromatin-bound Rb. Green arrows: cells that have inactivated both Rb and APC/C^CDH1; blue arrows: cells that have inactivated Rb but not APC/C^CDH1; orange arrows: cells that have inactivated APC/C^CDH1 but not Rb. Scale bar: 10 μm. One of $n = 2$ biological replicates.

both conditions, cells in G0 or early G1 are marked by low APC/C degron expression and low Rb phosphorylation signal; and cells in late G1, S, or G2 are marked by high APC/C degron expression and high phosphorylated Rb signal. With this knowledge, we stained for phospho-Rb (S807/S811) in the small intestinal crypts of mice expressing the APC/C degron reporter[43]. Strikingly, while many crypt cells have high phosphorylated Rb signal and low APC/C^CDH1 degron expression (marking cells with the canonical sequence of signaling order) (Fig. 5c, control condition, examples denoted by blue arrows and outlines), others have high APC/C degron expression and low Rb phosphorylation signal (marking cells with the reversed signaling sequence) (Fig. 5c, control condition, orange arrows and outlines).

To quantify these observations, we first used post-mitotic cells as a negative control to unbiasedly select thresholds for phosho-Rb and APC/C degron positive cells, and then we calculated the percentage of cells with high phospho-Rb signal, low APC/C degron signal or vice versa (Supplementary Fig. 8a). This analysis demonstrates that both order of events, corresponding to CDK4/6 and CDK2-initiated paths, can be found in crypt cells under

normal unperturbed conditions (Supplementary Fig. 8b, control condition). A critical control experiment that further strengthened this conclusion was systemic CDK4/6 inhibition in mice via oral administration of Palbociclib. CDK4/6 inhibition lowered the percentage of crypt cells with high phosphorylated Rb signal and low APC/C degron, but did not affect the cells with high APC/C degron signal and low Rb phosphorylation signal, consistent with our hypothesis that some crypt cells undergo CDK4/6 activity-independent cell-cycle entry (Fig. 5c; quantification: Supplementary Fig. 8b, CDK4/6 inhibitor condition). Due to technical reasons, we were not able to co-stain with total Rb; however, we note that Rb level does not decrease in MCF-10A cells after phosphorylation, suggesting that phospho-Rb can be used as an approximation for normalized phospho-Rb (Supplementary Fig. 8c). Based on this data, we conclude that cells in physiological settings can start the cell cycle by relying more heavily on cyclin D-CDK4/6 or cyclin E-CDK2.

**APC/C^CDH1 inactivation remains reversible until Rb inactivation in cells lacking CDK4/6 activity.** We next focused on the

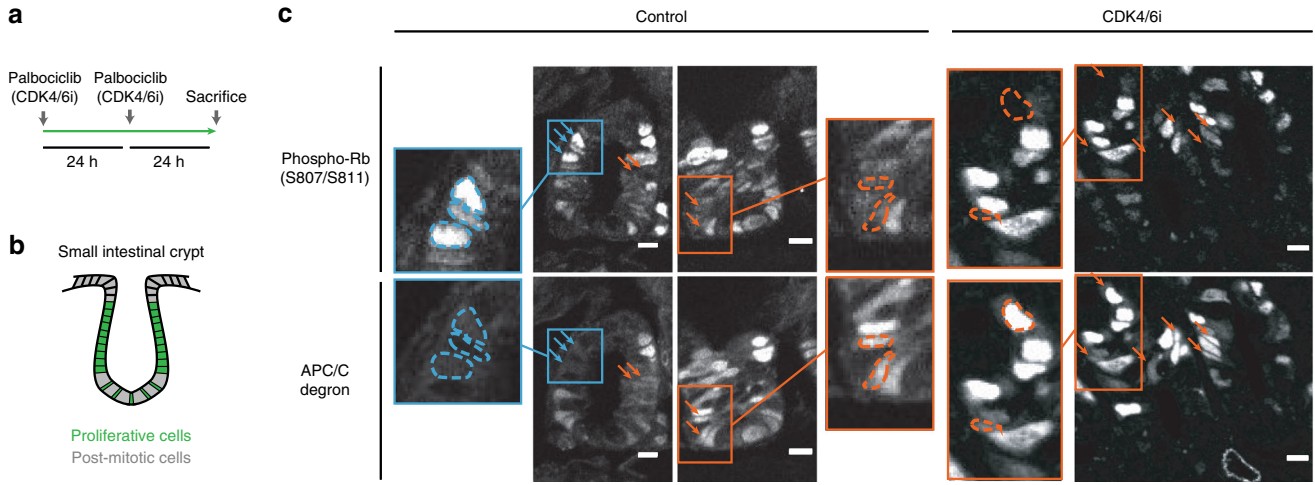

**Fig. 5 A fraction of small intestinal crypt cells have the unphosphorylated Rb and inactive APC/C$^{CDH1}$ characteristic of cells entering the cell cycle without CDK4/6 activity. a** Experiment setup for drugging mice. **b** Illustration of proliferative and post-mitotic cells in the small intestinal crypt. **c** Mice small intestinal crypts that were treated with vehicle or CDK4/6 inhibitor. The cells express the APC/C activity reporter and were stained for phospho-Rb (S807/S811). Blue arrows and outlines denote example cells with high phospho-Rb and low APC/C degron signal. Orange arrows and outlines denote cells with high APC/C degron and low phospho-Rb signal. Crypts are oriented in U-shapes. Scale bar = 10 μm. Representative images from $n = 5$ mice for control and $n = 6$ mice for CDK4/6 inhibitor.

question of when cells without CDK4/6 activity start S phase relative to the different signaling events in G1. The reversed order of Rb and APC/C$^{CDH1}$ inactivation in CDK4/6-inhibited cells suggest that the late Rb inactivation might be a new rate-limiting step in G1 progression with cells now waiting for Rb instead of APC/C$^{CDH1}$ inactivation for commitment. First, by plotting phospho-Rb (S807/S811) signal against incorporated EdU signal, we confirmed that Rb is phosphorylated prior to DNA replication (Fig. 6a). We then found that in both mitogen released and asynchronously cycling CDK4/6-inhibited cells, Rb is phosphorylated ≥4 h after APC/C$^{CDH1}$ inactivation, and DNA is synthesized ≥6 h after APC/C$^{CDH1}$ inactivation as opposed to only 1 h after APC/C$^{CDH1}$ inactivation in CDK4/6-dependent cells (Fig. 6b; cycling cells: Supplementary Fig. 9a). The long delay from APC/C$^{CDH1}$ inactivation to S phase in cells without CDK4/6 activity suggests that cells wait for Rb inactivation instead of APC/C$^{CDH1}$ inactivation before they can start replicating DNA.

Given this long delay between APC/C$^{CDH1}$ inactivation and S phase, we proceeded to ask when cells without CDK4/6 activity irreversibly commit to S-phase entry, a necessary step for maintaining the proper order of cell-cycle phases and genome integrity[44–46]. Irreversible APC/C$^{CDH1}$ inactivation, which allows cells to permanently accumulate necessary S phase proteins such as cyclin A, marks the point of no return in G1 in cells with CDK4/6 activity[14]. In CDK4/6-initiated cell-cycle entry, this commitment point occurs immediately after APC/C$^{CDH1}$ inactivation[14] when Emi1, an APC/C$^{CDH1}$ inhibitor and E2F target, facilitates irreversible APC/C$^{CDH1}$ inactivation via a double negative feedback loop (Fig. 6c)[14,15]. Without this irreversible inactivation by Emi1, genome fidelity can be compromised[13,47,48]. In CDK4/6-inhibited cells, however, E2F-mediated Emi1 transcription is delayed similar to that of E2F1 (Supplementary Fig. 9b), and initial APC/C$^{CDH1}$ inactivation kinetics is slower, characteristic of cells lacking Emi1 (Supplementary Fig. 9c, d)[14]. This leads to the hypothesis that irreversible APC/C$^{CDH1}$ inactivation occurs not at the onset of APC/C$^{CDH1}$ inactivation but only later. To directly test for reversibility of APC/C$^{CDH1}$ inactivation, we again used the APC/C degron sensor. Since the APC/C degron is constitutively expressed, if APC/C$^{CDH1}$ is off, the degron level will increase linearly, but if APC/C$^{CDH1}$ is turned

back on, the degron level will fall back down. To test for APC/C$^{CDH1}$ activity reversibility, we first used a CDK1/2 inhibitor, which (1) removes the original cyclin E-CDK2 activity input that turned off APC/C$^{CDH1}$ (a test for irreversibility), and (2) can simulate stress-mediated decrease of cyclin E/A-CDK activity[14,36]. Strikingly, unlike cells with active CDK4/6 (Fig. 6d, log scale included to aid degron visualization at low levels), cells without CDK4/6 activity reactivated APC/C$^{CDH1}$ after CDK1/2 inhibition, as seen by a reversal and drop of the APC/C degron signal, demonstrating that APC/C$^{CDH1}$ inactivation is reversible (Fig. 6e, gray box). APC/C$^{CDH1}$ inactivation only becomes irreversible several hours later, coinciding with Rb inactivation (Fig. 6e, right panels), and knockdown of Emi1 at this time point reactivates APC/C$^{CDH1}$ (Supplementary Fig. 9e, f). From these experiments, we conclude that APC/C$^{CDH1}$ inactivation is initially reversible in CDK4/6-inhibited cells, and that irreversible APC/C$^{CDH1}$ inactivation occurs several hours later, coinciding with the synthesis of Emi1 mediated by inactivation of Rb.

The CDK1/2 inhibitor experiment suggests that in cells without CDK4/6 activity, the point of no return in respect to stress may occur later as well. In CDK4/6-initiated cell-cycle entry, cells remain sensitive to stress until but not after APC/C$^{CDH1}$ inactivation (for hypertonic stress, Fig. 6f, g; for oxidative stress, Supplementary Fig. 9g, h), demonstrating that APC/C$^{CDH1}$ inactivation represents the stress commitment point[14]. To determine when CDK4/6-inhibited cells lose stress sensitivity, we applied acute stress at different times after APC/C$^{CDH1}$ inactivation. Markedly, in these asynchronously cycling cells, stress applied 1 h after APC/C degron detection still resulted in reactivation of APC/C$^{CDH1}$ (Fig. 6h and Supplementary Fig. 9i, gray boxes). We next determined if Rb inactivation coincides with a loss of stress sensitivity in the cyclin E-CDK2-initiated path by examining cells that were (1) treated with stress five hours after APC/C$^{CDH1}$ degron detection, and (2) had cyclin E/A-CDK activity > 0.8 at the time of treatment. The additional CDK activity gating (threshold determined from Fig. 3e) enriches for cells with inactivated Rb at the time of stress treatment. Applying these gating conditions, we found that APC/C$^{CDH1}$ inactivation indeed became irreversible in respect to stress (Fig. 6h; Supplementary Fig. 9i, right panels). Thus, in cells with minimal CDK4/6 activity,

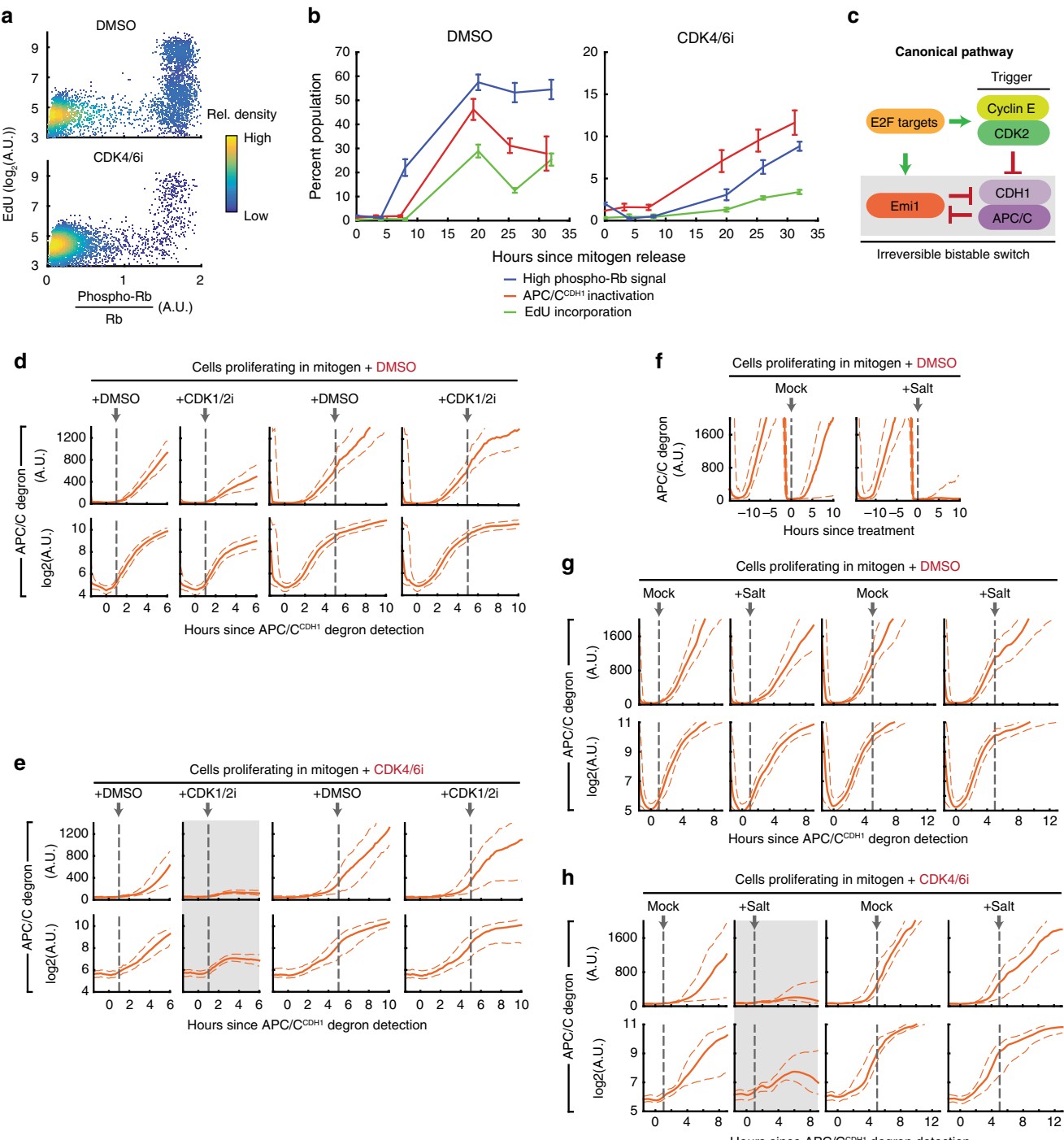

**Fig. 6 APC/C^CDH1 inactivation remains reversible until Rb inactivation in cells lacking CDK4/6 activity. a** Cells born into DMSO or 1 μM CDK4/6 inhibitor were treated with 10 μM EdU for 15 min, and then assayed for EdU and phospho-Rb(S807/S811) (5000 cells plotted, 1 of *n* = 3 biological replicates). **b** Mitogen-released cells treated with DMSO or CDK4/6 inhibitor were measured for percent cells with high phospho-Rb(S807/S811) signal, high APC/C degron, and high EdU signal (after 15 min of 10 μM EdU treatment). Mean ± SEM from three biological replicates. **c** Emi1 ensures irreversible APC/C^CDH1 inactivation. **d**–**h** Cells born into DMSO or CDK4/6 inhibitor and have inactivated APC/C^CDH1 (with the exception of (**f**), where it is just cells born into DMSO or CDK4/6 inhibitor) were treated with 3 μM CDK1/2i (**d**, **e**) or an extra 100 mM salt (**f**–**h**). Dashed lines denote 25th and 75th percentile and bottom row panels are plotted in log_2 scale for better visualization. APC/C degron detected via signal over background. Gray boxes denote conditions where cells reactivated APC/C^CDH1. *n* = 2 biological replicates; *n* > 30 cells per condition. In **g**, **h**, to enrich for Rb inactivated cells in the 5 h post APC/C^CDH1 degron detection conditions, only cells with cyclin E/A-CDK >0.8 right before treatment were analyzed (threshold determined from Fig. 3e).

the commitment point, where APC/C$^{CDH1}$ inactivation becomes irreversible and cells become insensitive to stress, coincides with Rb inactivation many hours after APC/C$^{CDH1}$ inactivation.

We note that cells on average lose sensitivity to stress after losing sensitivity to mitogen (the point of mitogen insensitivity is commonly referred to as the restriction point)[13,14,49]. Both of these commitment points were recently shown to be more gradual processes than previously thought[7,50]. Cells that rely on CDK4/6 activity to proliferate, however, are insensitive to mitogen removal and exogenous stress by the time APC/C$^{CDH1}$ is inactivated (Supplementary Fig. 10, DMSO condition; cells that turned APC/C$^{CDH1}$ back on are marked in orange)[7,13,14,50]. In cells without CDK4/6 activity, however, cells remain sensitive to mitogen withdrawal after the initial APC/C$^{CDH1}$ inactivation, and mitogen sensitivity is then lost gradually over time (Supplementary Fig. 10, CDK4/6i condition). This data demonstrates that not only stress, but also mitogen sensitivity is lost later in cells entering the cell cycle without CDK4/6 activity.

Finally, given the unexpected difference in signaling order and commitment process in cells proliferating without CDK4/6 activity, we tested for potential cell-cycle defects. We found that these cells exhibit similarly low S/G2 phase DNA damage as measured by γH2A.X and 53BP1 signals (Supplementary Fig. 11a, b). Based on these results, it is unlikely that the S and G2 phases of the cell cycle are severely affected by the reversed order of Rb and APC/C$^{CDH1}$ inactivation or the change in the point of no return. Combined with the small intestinal crypt data, our data demonstrates that the cyclin E-CDK2-initiated path is a viable path for cell-cycle entry.

## Discussion

Our work provides mechanistic insights to seminal genetic work which established that cells can proliferate without cyclin D-CDK4/6 activity[16,51]. In the case of ablated cyclin Ds, the adaptability is due to activation of cyclin E-CDK2, though how cyclin E-CDK2 is activated without cyclin D-CDK4/6 activity was unknown in wild-type cells (genetic mutations may not reflect acute depletions[52,53]). Our study delineates sequential molecular events in the CDK2-initiated path that are distinct from the CDK4/6-initiated path (Fig. 7). Specifically, we show that in the CDK2-initiated path, the initial activation of cyclin E-CDK2 is delayed, greatly varies between cells in the same population, and is less efficient compared to the CDK4/6-initiated path. In addition, we also find that without CDK4/6 activity, the rate of increase in CDK2 activity is slower and often fluctuates over time, demonstrating a previously unknown role for cyclin D-CDK4/6 in conferring persistence to cyclin E-CDK2. In the CDK2-initiated path, the initial increase in CDK2 activity is also independent of E2F target gene expression and requires c-Myc activity. This cyclin E-CDK2-initiated path may have main or backup roles in regulating proliferation in a variety of biological contexts. For example, due to its much longer G1, the cyclin E-CDK2-initiated path may play roles in controlling tissue replacement rates. In addition, longer G1 may also be important in regulating cell differentiation[54].

Most strikingly, we show that the order between Rb and APC/C$^{CDH1}$ inactivation is reversed in the CDK2-initiated path with APC/C inactivation occurring approximately 4 h before instead of after Rb phosphorylation. This, to our knowledge, is the first demonstration of G1 temporal signaling plasticity. We also observed this reversed signaling order in different cell types, including crypt cells in vivo. Finally, we show that in the CDK2-initiated path, the point of no return coincides with Rb inactivation rather than APC/C$^{CDH1}$ inactivation as in the CDK4/6-initiated path. It is interesting that in both MCF-10A and BJ5-ta cells growing in culture, some cells exhibit characteristics of the CDK2-initiated path (Fig. 3c, d, DMSO condition where CDK2 is activated without Rb hyperphosphorylation). While different cell types are likely to have different percentages of cells utilizing different paths, it is also possible that cell type-specific entry may reflect a combination of CDK4/6 and CDK2-initiated paths that can cooperate with one another to inactivate Rb and APC/C$^{CDH1}$.

Given the increasing number of usage for CDK4/6 inhibitors in cancer treatment and the potential of cyclin E-CDK2 mediated drug resistance[22–25], our study also has therapeutic relevance. Our data showed that this alternate cell-cycle entry mechanism is often marked by fluctuating cyclin E-CDK2 activity and is more sensitive to stress (Fig. 2b, d). Furthermore, early inactivation of APC/C$^{CDH1}$ in the CDK2-initiated path facilitates Rb inactivation. Both of these unique features may offer opportunities for exploiting synthetic lethality in cancers. Finally, when considering that (1) >10 billion normal cells need to be replaced every day[55,56] and (2) CDK4/6 inhibitors do not have toxicity issues common among other CDK inhibitors (the side effects of CDK4/6i are primarily hematopoietic)[57], the use of this alternate mechanism may explain why most normal tissues tolerate CDK4/6 inhibitors during cancer therapy.

## Methods

**General experimental setup**. Cells were seeded to ensure 30–90% confluency throughout the experiment in 96-well plates. In the majority of the experiments, the same numbers of cells were plated for all conditions (one exception being the multigenerational experiment that involved 72 h of live imaging, where the DMSO condition started off with 75% of the cells in the CDK4/6 inhibitor condition). In mitogen starvation experiments, cells were left in the starvation media for 48 h after being washed at least once. Cells were then mitogen released with and without Palbociclib and refreshed every 24 h unless indicated otherwise.

**Cell lines**. All cell lines were acquired from ATCC unless noted otherwise. Cells were always grown in 37 °C and 5% CO$_2$ and between 30 and 80% confluency. Cells were provided fresh media at least once every three days. Dulbecco Modified Eagle medium (DMEM) and DMEM/F12 were acquired from ThermoFisher Scientific. MCF-10A (ATCC, #CRL-10317, human female) were cultured in phenol red-free DMEM/F12 supplemented with 5% horse serum, 20 ng/mL EGF, 10 μg/mL insulin, 500 μg/mL hydrocortisone, and 100 ng/mL cholera toxin. Starvation media consisted of the same growth media but instead of horse serum, insulin, and

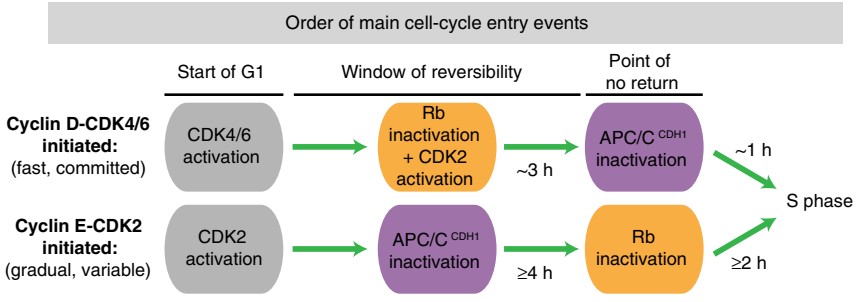

**Fig. 7 Working model.** Distinct regulatory programs in cells entering the cell cycle with and without CDK4/6 activity.

EGF, 0.3% bovine serum albumin (BSA) is added. Re-suspension media (used to inactivate trypsin) consisted of DMEM/F12 and 20% horse serum (protocol from the laboratory of Joan Brugge). All MCF-10A experiments were conducted on cells that went through <15 passages (passage 1 being receipt from ATCC), and cell line identity was confirmed through RNA-seq. BJ-5ta (ATCC, #CRL-4001, human male) were cultured in DMEM plus 10% fetal bovine serum (FBS), 20% Medium 199 (ThermoFisher), and 0.02 mg/ml hygromycin B. Starvation media consisted of DMEM, 0.1% BSA, and 0.02 mg/ml hygromycin B. HS68 (ATCC, #CRL-1635, human male), passage 10, and WI-38 (ATCC, #CCL-75, human female), passage 5, were cultured in DMEM plus 10% FBS. MDA-MB-435 (ATCC, #HTB-129, human female) were cultured in Leibovitz's L-15 medium (ATCC) plus 0.01 mg/mL bovine insulin, 0.01 mg/mL glutathione, and 10% FBS. Colo-205 (ATCC, #CCL-222, human male) was cultured in RPMI-1640 (ThermoFisher) with 10% FBS. U2OS (human female), acquired from the laboratory of Karlene Cimprich, and HeLa (ATCC, #CCL-2, human female) were cultured in DMEM plus 10% FBS. Panc 08.13 (ATCC, #CRL-2551, human male) were cultured in RPMI-1640 (Thermo-Fisher) plus 10 Units/mL human recombinant insulin, and 15% FBS. SCC-9 (ATCC, #CRL-1629, human male) were cultured in 1:1 mixture of DMEM and Ham's F12 containing 1.2 g/L sodium bicarbonate, 2.5 mM L-glutamine, 15 mM HEPES and 0.5 mM sodium pyruvate plus 400 ng/mL hydrocortisone and 10% FBS (protocol from ATCC).

**Stably transfected cell lines**. All constructs were introduced into cells by lentiviral transduction. CSII-pEF-H2B-mTurquoise, CSII-pEF-DHB(aa994–1087)-mVenus, CSII-pEF-DHB(aa994–1087, serine to alanine mutant)-mCherry and CSII-pEF-mCherry-Geminin(aa1-110) were described previously[14,32,33]. Transduced cells were sorted on a Becton Dickinson Influx to obtain populations expressing the desired fluorescent reporters. DHFR-Myc[58] was cloned into pCru5-IRES-puro and selected with 1 µg/ml puromycin (until the mock transfected cells all died).

**Chemical inhibitors, DNA damage, mitogen withdrawal, salt, H₂O₂, EdU treatment**. The inhibitors used in this study were: CDK1 inhibitor RO3306 at 10 µM (Sigma-Aldrich SML0569), CDK4/6 inhibitor PD0332991 (Palbociclib) at 1 µM unless indicated otherwise (Selleck Chemicals S1116), CDK4/6 inhibitor LY2835219 (Abemaciclib) at 3 µM (Selleck Chemicals S7158), CDK4/6 inhibitor LEE011 (Ribociclib) at 9 µM (Selleck Chemicals S7440), and CDK1/2 inhibitor CAS 443798-55-8 at 3 µM (EMD Biosciences #217714).

In experiments involving CDK4/6 inhibitor, cells were treated long term. In Rb phosphorylation experiments involving CDK1i and CDK1/2i, cells were treated for the last 20 min prior to fixation. In APC degron experiments involving CDK1/2i, cells were treated with CDK1/2i for 4 h. In experiments involving neocarzinostatin and aphidicolin, the concentrations and durations are indicated in the legends and figures. For mitogen withdrawal experiments, cells were washed three times with starvation media, and then left in the starvation media for the remainder of the experiment. In experiments where hypertonic conditions are created, a 3 M stock of salt in water (30x) is spiked in cells, the final media osmolality is ~440 mOsm/kg. For oxidative stress, 200 µM of H₂O₂ was applied. For assays involving EdU staining, cells were treated with 10 µM EdU for ~15 min or 100 µM for 5 min prior to fixation unless noted otherwise (cells in Fig. 1c and Supplementary Fig. 2d were treated with 100 nM or 1 µM EdU for 24 h).

**siRNA transfection**. Cells were transfected using Dharmafect 1 (ThermoFisher) according to the manufacturer's instructions. The following Dharmacon siRNAs were used: control siRNA (nontargeting #2), siGenome Human Myc (4609) set of four, Human siGenome Human CCNE1/Cyclin E1 (3213-09), siGenome Human CDKN1A/p21 (1026) set of four, siGenome FBXO5/Emi1 (12434-01, 02, 03), siGenome FZR1/Cdh1 (MQ-015377) set of four. siRNAs were used at a final concentration of 20 nM. Cells were transfected 30–40 h after starvation (the siRNAs were diluted in starvation media) and washed out when released with growth factors.

**Western blot**. Cells were washed once with cold phosphate-buffered saline (PBS), and then lysed using 10 mM Tris pH 7.5, 2 mM EDTA, and 1% sodium dodecyl sulfate. Harvested materials were then sheared using a 25 G (0.5 mm) syringe, and denatured at 90 °C for 5 min. Samples were ran on tris-glycine gels. Membranes were blocked in TBST + 5% milk, and then treated with primary antibodies, horseradish peroxidase-conjugated secondary antibodies, and chemiluminescent substrate prior to exposure to film.

Antibodies used: Cyclin E (Santa Cruz, sc-247, 1:400), p21 (Cell Signaling, #2947, 1:1000), Cdh1/Fzr1 (Abcam, AB89535 1:1000), c-Myc (Cell Signaling, #5605, 1:1000), Emi1/Fbxo5 (Invitrogen, 37-6600, 1:500), GAPDH (Cell Signaling, #5174, 1:5000)

**c-Myc induction**. Cells stably expressing DHFR-Myc were treated with 10 µM Trimethoprim (TMP) at the time of release. TMP was refreshed every 24 h along with the CDK4/6 inhibitor.

**Time-lapse microscopy**. Images were taken on an IXMicro microscope (Molecular Devices). 10× objective (0.3 N.A.) used for live-cell imaging and fixed-cell

immunofluorescence imaging. 20× objective (0.75 N.A.) with 2-by-2 pixel binning for imaging RNA FISH staining. For live-cell imaging, images were taken every 12 min, and total light exposure time was kept under 600 ms for each time point. Cells were imaged in a humidified, 37 °C chamber in 5% CO₂.

**Tissue culture immunofluorescence**. Tissue culture cells were fixed in 4% par-aformaldehyde for 10 min, then blocked in PBS containing 10% FBS, 1% BSA, 0.1% TX-100, and 0.01% NaN₃ for 1 h at room temperature, and then stained overnight in 4 °C with antibodies for immunofluorescence (in blocking buffer). Alexa Fluor secondary goat antibodies (ThermoFisher) were applied 1:2000 for 1 h at room temperature in blocking buffer, and then added with Hoechst 33342 (Thermo-Fisher) at 1:10,000 for ten minutes at room temperature. Cells are stored in PBS when imaged. In the case where cells expressed fluorescent reporters that limited the use of fluorophores, cells were chemically bleached after fixation[59]. In the case of cyclin E1 staining, ice-cold methanol was applied for ten minutes instead of paraformaldehyde for fixation. In experiments where incorporated EdU signal is measured, the Click reaction is performed after blocking (if photobleaching is required, Click reaction also is performed after) following manufacturer's protocol (Invitrogen, #C10269).

Antibodies used Rb(p-S807/811) (Cell Signaling, #8516, 1:2500), Rb (Cell Signaling, #9309, 1:1000), Cyclin E (Santa Cruz, sc-247, 1:400), p21 (Cell Signaling, #2947, 1:250), geminin (Human Atlas, HPA049977, 1:500), c-Myc (Cell Signaling, #5605, 1:800), γH2A.X(S139) (Cell Signaling, #2577 1:800), and 53BP1 (EMD Millipore MAB3802, 1:1000).

**RNA FISH**. RNA in situ hybridization was carried out using the Affymetrix Quantigene ViewRNA ISH cell assay as specified in the user manual right after fixation (precedes Click reaction or immunofluorescence). Pre-made probes were designed to target E2F1 and Emi1. RNA in situ hybridization was carried out using the Affymetrix Quantigene ViewRNA ISH cell assay kit following manufacture's protocol.

**Pre-extraction**. Rb pre-extraction protocol was described previously[40], but rather than trypsinizing and FACS, pre-extraction and fixation were performed with the 96-well plate on an ice block. Antibody used: Rb (BD Biosciences #554136, 1:250). When EdU needs to be incorporated, the incorporation precedes the permeabilization.

**Treatment, isolation, and immunofluorescence of mice small intestinal crypts**. Fucci2a mice (expression of mCherry-hCdt1(30–120)-T2A-Venus-hGem(1–110), RDB13080, RIKEN)[43] were made by crossing lox-stop-lox Fucci2a mice (Gt (ROSA)26Sor tm1(Fucci2aR)Jkn) with EIIa-Cre mice (B6.FVB-Tg (EIIa-cre) C5379Lmgd/J). EIIa-Cre mice were a gift from Mitinori Saitou, Kyoto University, Kyoto, Japan. The mice were dosed with 150 mg/kg of Palbociclib (30 mg/mL in sodium lactate buffer). Palbociclib was prepared fresh and administered orally via gavage. Mice were re-dosed after 24 h and sacrificed after 48 h. Immediately after sacrifice, the proximal, central, and distal regions of the small intestine were collected and fixed in 4% paraformaldehyde overnight in 4 °C. Tissues were then washed and treated with 12% sucrose for 2 h, 15% sucrose for 2 h, and then 18% sucrose overnight (all sucrose steps conducted in 4 °C). Tissues were then embedded in optimal cutting temperature compound (Sakura Finetek Japan), and cryosectioned at 6 µm thickness. Samples were then air-dried, washed, permeabilized/blocked (0.1% Triton X-100 and 1% BSA in TBST for 1 h in room temperature), and treated with primary antibody Rb(p-S807/811) (Cell Signaling, #8516, 1:1000) in blocking buffer at 4 °C overnight. Alexa Fluor secondary goat anti-rabbit 647 (ThermoFisher) was applied at 1:2000 for 1 h at room temperature in blocking buffer, and images were acquired immediately after using an FV1000/IX83 confocal microscope (Olympus). To avoid bleedthrough from the mCherry signal, phospho-Rb signal was collected at wavelengths of 690–790 nm. Both female and male mice were used, and the mice age range from 6 to 11 weeks. The animal protocols were reviewed and approved by the Animal Care and Use Committee of Kyoto University Graduate School of Medicine (No. 18086).

**Quantification and statistical analysis**. Image analyses for live-cell and fix-cell assays have been described previously[7,14]. In short, images were first flatfield corrected (illumination bias determined by pooling background areas from multiple wells from the same imaging session), and then background subtracted (background determined locally for APC/C degron signal, and globally for other signals). Nuclei were segmented using Hoechst or H2B-mTurquoise. Cytoplasmic signal in each cell was estimated via a ring mask with inner radius of 2 µm and an outer radius of at most 10 µm outside of the nuclear mask (for mRNA FISH, at most 50 µm) without overlapping neighboring cells' nuclear or cytoplasmic ring masks. Median signal values were obtained for all signals except for DNA content (calculated by summing total nuclear Hoechst intensity), DHB cytoplasmic signal (calculated from median intensity of pixels that exceed global background values), mRNA FISH (pixel count in both cytoplasmic and nuclear region after top-hat filtering and thresholding on intensity), and γH2A.X(S139)/53BP1 (same quantification method as mRNA FISH except only using nuclear mask). A cell is tracked by identifying its nearest future neighbor that matches its total H2B-mTurquoise fluorescence. Mitosis is called when both of the two nearest future neighbors have

H2B signals that are 45–55% of the original cell. Derivation of APC/C$^{CDH1}$ activity has been described previously[14]. Briefly, we assumed constant reporter synthesis due to expression under a constitutive promoter, and derived the reporter synthesis rate ($k_{syn}$) from the increase in APC/C-degron signal during steady state. We then calculated APC/C activity using the equation

$$\text{APC/C activity} = \frac{k_{syn} - \frac{d}{dt}(\text{APC/C degron})}{[\text{APC/C degron}]}$$

In Supplementary Figs. 5a and 9a, and Fig. 6b, the time of APC/C$^{CDH1}$ inactivation is moved earlier by 48 min to adjust for the mCherry maturation time[14]. APC/C degron detection was determined via signal over background degron signal. Information about replicates and error bars can be found in the figure legends. In box plots, the central line indicates median, the edges of the box denote the quartiles, and the whiskers extend to the farthest points that are not outliers (defined as 1.5 times the interquartile range away from the box edges). Automated-pipeline scripts performed all analyses. In analyses where only certain populations of cells are analyzed (for example, G0/G1 cells), the gating criteria are described in the figures, figure legends, or results section. In experiments where significance was derived, unpaired two-sample $t$ tests were performed. In Supplementary Fig. 9a, the time denoted by the gray box is estimated.

Experiments with MCF-10A and BJ-5ta are excluded when cells are of suboptimal confluency and/or unhealthy (<30% of control cells in S/G2 24 h after mitogen release or when asynchronously cycling).

Crypts were selected for analysis based on the following criteria: (i) contains at least ten cells expressing the sensors, (ii) shaped like the letter U, and (iii) contains post-mitotic cells as negative controls in the bottom of the crypt. Cells were segmented manually in ImageJ (by merging the three nuclear channels: geminin-fragment, Cdt1-fragment, and phospho-Rb) and median nuclear intensities were then calculated. To calculate the thresholds for determining APC/C-degron positive and phospho-Rb-positive cells, the signals of post-mitotic cells from the same crypt were averaged and multiplied by a small factor. The same unbiased approach for threshold determination is applied to every image.

**Reporting summary**. Further information on research design is available in the Nature Research Reporting Summary linked to this article.

## Data availability
Source data for uncropped gel images and data underlying reported averages are provided as a Source Data file. Additional data are available from the corresponding author upon reasonable request. Source data are provided with this paper.

## Code availability
The code for the image analysis pipeline is available at https://github.com/scappell/Cell_tracking. Additional modified code is available from the corresponding author upon reasonable request.

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

## Acknowledgements

We thank M. Köberlin, L. Pack, and A. Cunningham for critical reading of the paper. We thank past and present members of the Meyer lab for reagents and discussions, Karlene Cimprich, James Ferrell, and Julien Sage for helpful discussions, and the Stanford Shared FACS facility for cell sorting. We also thank the laboratories of Richard Mort, Steven Cappell, Karlene Cimprich, and Thomas Wandless for providing the Fucci2a mice, constructs, cell lines, and chemicals. C.L. was supported by the NSF Graduate Research Fellowship, L.H.D. was supported by NIH training grants T32GM007365 and F30AG060634, and T.M. was supported by NIH grants R35 GM127026; Y.K., K.T., and M.M. were supported by JSPS 15H05949 "Resonance Bio" and 16H06280 "ABiS". Further information and requests for resources and reagents should be directed to and will be fulfilled by the lead contact, Tobias Meyer (tobias1@stanford.edu).

## Author contributions

Conceptualization: C.L. and T.M.; Methodology: C.L., and M.C.; Software: C.L. and M.C.; Formal analysis: C.L.; Investigation: C.L., Y.K., M.C., L.H.D., and Y.F.; Resources: C.L., H.W.Y., K.T., M.M., and T.M.; Writing: C.L., and T.M.; Visualization: C.L.; Supervision: T.M.; Funding acquisition: C.L., K.T., M.M., and T.M.

## Competing interests

The authors declare no competing interests.
