## [Peer Review File · Nature Communications]

Reviewers' comments:

Reviewer #1 (Remarks to the Author):

The authors have carried out a sophisticated analysis of the regulatory system that drives entry into the mammalian cell cycle. The work clarifies the role of specific components and reveals how the system adapts in response to removal of certain key components. These results will be of interest to the many readers studying the control of cell proliferation in normal and tumor cells.

The current view of the cell cycle entry system is roughly the following: Mitogens → Myc → cyclin D-Cdk4/6 → Rb phosphorylation → E2F activation → expression of cyclin E, A, Emi1 → Cdh1 inactivation → cyclin A stabilization → S phase. This system includes numerous single and double feedbacks (for example, cyclin E-Cdk2 can also phosphorylate Rb, leading to positive feedback; double negative feedback is provided by mutual antagonism between Cdh1 and Emi1 or cyclin A). This is therefore a complex and robust regulatory switch that withstands inactivation of certain key molecules. For example, it is known that cyclin D-Cdk4/6 knockouts are not lethal, and cyclin E-Cdk2 promotes cell cycle entry under these conditions.

The current paper provides interesting and important new molecular details about how the system responds to the removal of cyclin D-Cdk4/6 activity, and how cyclin E-Cdk2 provides compensation. The authors use acute chemical inhibition of Cdk4/6 activity to explore the mechanisms by which the cell adapts to the loss of this kinase. This approach is superior to the previous approach of gene knockouts, allowing more detailed analysis of immediate effects. In the absence of cyclin D-Cdk4/6 activity, cyclin E gradually accumulates, presumably because it is a Myc target. As it accumulates, cyclin E-Cdk2 phosphorylates two targets that are relevant here: Rb (leading to E2F activation and further cyclin E/A production) and Cdh1 (leading to APC inactivation and stabilization of cyclin A). In a normal cell, cyclin D-Cdk4/6 is a specific Rb kinase and initiates Rb phosphorylation before cyclin E-Cdk2 activity accumulates; thus, Rb is phosphorylated well before Cdh1 is inactivated. In the absence of cyclin D-Cdk4/6, however, the authors show that the order of these phosphorylation events is reversed: Cdh1 is phosphorylated slightly before Rb. This defect in the sequence of events presumably results from the fact that cyclin E-Cdk2 is now the kinase primarily responsible for both phosphorylation events. It seems likely that Cdh1 is a better cyclin E-Cdk2 substrate and is modified first.

This reversal of the normal order of regulatory events leads to a new scheme in which Rb phosphorylation, rather than Cdh1 inactivation, is now the final step to cell cycle commitment. The work therefore suggests that both Rb phosphorylation (=E2F activation) and Cdh1 inactivation (=cyclin A stabilization) are essential for cell cycle entry, but their precise order is not critical. Previous work (including work from these authors) has argued that Rb phosphorylation represents the critical event for cell cycle commitment, but the current paper provides a more nuanced and interesting view: that two major regulatory events together drive commitment.

Reviewer #2 (Remarks to the Author):

The results of this paper are over-interpreted. The title “A parallel cell cycle entry pathway with inverted G1 signalling and alternate point of no return” sounds like a dramatic discovery, but amounts to hype as it is not strongly supported by the data. The genetic deletion of CDK4 and/or CDK6 in various metazoan systems has shown long ago that these kinases are not required for entry into the cell cycle, although they enhance it. So the discovery that CDK4/6 inhibition slows but does not prevent entry into the cell cycle is not novel. I failed to grasp the reasoning behind the need to invoke the existence of two alternate “pathways” rather than simply a single mechanism which is slower when CDK4/6 activity is inhibited (ie redundant CDK activity, resulting in both Rb inactivation and CDH1 inactivation, at slightly different kinetics depending on whether CDK1/2/4/6 are active or only CDK1/2). This might sound less exciting but to me reflects equally well the overall conclusions from the experiments, which are complex. Indeed, the single cell approach, while elegant, requires some quite complex gating, is not always easy to follow, and appears over-interpreted in places. The single cell reporters have their limitations: for example, the CDK2 activity reporter represents the ratio of the sum of CDK1/2 activity with respect to the phosphatase activity, rather than the claimed cyclin E-CDK2 activity. In some instances a population approach would allow to reinforce some conclusions. Below I present my analysis of the experiments.

1. In the experiments, starting with that in figure 1, the concentration of palbociclib used is 1 μ M. In these conditions, 10% of cells complete S-phase, whereas 30% progress through G1 (as assessed by APC/C inactivation) but with a strong delay compared to controls. What happens with lower or higher doses (100nM, 10 μ M)? The concentration used should be justified and evidence presented to support the claim that no residual CDK4/6 activity exists. This is critical for the interpretation of essentially all the experiments in the paper. Showing that 3 μ M rather than 1 μ M causes no further inhibition of S-phase entry is not appropriate: it says nothing about cycD-CDK4/6 activity, since S-phase entry is the consequence of normal CDK4/6 activity and is what is being measured. It would be thus be a circular argument to claim that this proves that all CDK4/6 activity is inhibited. The authors themselves suggest, for example, that cells can go into S-phase with a low or no CDK4/6 activity. A direct readout for CDK4/6 activity would be essential; for example, a blot analysis of monophosphorylated Rb species.

2. From the experiment in figure 1, the authors conclude that the results reveal an alternative pathway. There is no evidence for this.

3. Since, in Fig 2b, c, the total percentage of cells does not add up to 100, presumably the grey traces are the rest (this is not mentioned in the legend). Thus, even without CDK4/6i, it appears that 30% of cells do not activate CDKs. With CDK4/6i, this percentage goes up to 70%. So in both cases many cells do not use this “alternate pathway” – or, perhaps, these cells fail to reach a threshold overall activity required for S-phase entry.

4. I’m not convinced that the fluctuations measured by imaging genuinely reflect fluctuating CDK activity rather than some technical noise. This cannot simply be assumed without showing some other evidence for it.

5. In fig 2c, the siRNA control cells have 14.5 high and 20.9% fluctuating CDK reporter activity, yet in

fig 2b the corresponding numbers are 34.1% and 13.1%, respectively. Is this just biological variability, or does the siRNA control affect the cell cycle on its own? Either way, what can we say about this variability?

6. There are no WB controls shown for the efficiency of siRNA-mediated knockdown in any experiment.

7. In fig 2e, it is not explained how c-Myc levels were quantified and the original data is not shown, so how the authors conclude that a 50% increase in c-Myc causes the observed effects is unclear.

8. I don't understand the conclusion that "c-Myc and cyclin D-CDK4/6 activities are driving forces that can promote cell-cycle entry through two different paths – one with slowed and gradual CDK2 activation, and one with fast and persistent CDK2 activation". Why should the different kinetics of CDK2 activation when CDK4/6 or Myc are manipulated necessarily reflect different pathways, rather than simply a less active pathway? Besides, c-Myc is a target gene controlled by cyclin D-CDK4/6 activity, so imagining what these pathways would look like is difficult.

9. During the description of the experiment shown in figure 3, the Narasimha paper (eLife 2014, cited ref 37) is presented in conjunction with data on Rb S807/11 phosphorylation, to support the argument that with CDK4/6 inhibition, cyclin E-Cdk2 is upregulated by c-Myc. I am not convinced by this experiment, for several reasons. First, the data tend to suggest that CDK4/6 can monophosphorylate Rb on S807/811, as expected from the Narasimha paper. Thus, S807/11ph may not distinguish between Rb mono and hyperphosphorylation. Second, no data on cyclin E-Cdk2 activity is presented, the only data shown are single cell traces of DHB cytoplasmic/nuclear ratio. This perhaps corresponds to Cyclin E-CDK2 activity in normal circumstances (or rather, the ratio of CDK1/2 activity to protein phosphatase activity), but this may not be the case when CDK4/6 is inhibited. Rather than doing a single cell analysis, a population analysis of CDK2-cyclin E activity would resolve this issue. The advantage of the single cell approach is not obvious here.

10. The experiment described in figure 4 is very hard to understand. First, if cells are analysed at 32h after CDK4/6 inhibition, yet only 20h after control (DMSO) treatment, this surely means that APC/C inactivation is much later than without CDK4/6 inhibition? So the starting point is different. Second, it cannot be taken for granted, without generating a circular argument, that the level of degron expression can be converted into time after APC/C inactivation both in control and CDK4/6i conditions, because the reporter might be transcribed/translated at different rates in the latter. Even assuming all is OK, it could simply be that APC/C is inactivated at lower CDK 1/2 levels than are required to hyperphosphorylate Rb, whereas everything occurs more or less simultaneously when CDK4 and 6 are active. I don't understand the conclusion of an "inversion" of these events, much less "inverted G1 signalling" or what the real significance of this observation is.

11. It gets even more confusing in Fig 4b, where E2F1 transcripts are analysed, since it is stated that the way the experiment was done was to gate cells with high CDK1/2 activity but low APC/C degron. Yet in Extended data Fig 5a it is shown that APC/C activity is switched off at similar CDK1/2 activity levels whether CDK4/6 are active or inhibited. So in high CDK1/2 activity, APC/C degron levels should be high.

12. The in vivo analysis in fig 4f is not very convincing, as the interpretation is based on a small proportion of cells that are very hard to see (nuclei are not marked). It may also be an issue that after hyperphosphorylation, Rb protein itself may be decreased, and in these IF experiments there is no indication of the ratio of phospho Rb to total Rb.

13. Fig 5f-h is even harder to understand. The explanation is telegraphic, to say the least, about what was actually done, how the data tell us anything about Rb inactivation (which was not assessed as far as I can tell), and how commitment was defined. It looks suspiciously like a circular argument: commitment being the point beyond which APC/C is no longer reactivated after application of salt. Yet that is what is being measured to conclude this.

Other points:

1. In the introduction, the current model for cell cycle entry is carefully presented, justifying the study; but this is nevertheless somewhat misleading. It is said that cyclin D-CDK4/6 complexes lead to Rb inactivation. Indeed, it was previously thought that CycD-CDK4/6 progressively phosphorylate and inactivate Rb, thus de-repressing cyclin E transcription, triggering a positive feedback loop since cycE-CDK2 complexes complete inactivation of Rb. However, more recent work by the Dowdy lab showed, in non-transformed cells, that this model is incorrect; instead, cycD-CDK4/6 complexes mono-phosphorylate Rb, and this monophosphorylated species is active as a transcriptional repressor; Rb is then quantitatively hyper-phosphorylated, and inactivated, by CycE complexes later. This model should be presented in the introduction. It is rather presented later in the results section (see above, discussion of experiment shown in figure 3) but how the current results fit with this model is not discussed.

2. The conclusion of the inversion of the order of CDH1 inactivation and Rb inactivation in the absence of CDK4/6 activity possibly reflects specificity of CDK4/6 complexes for CDH1 itself, as shown for homologues in nematodes by the Van den Heuvel lab (evidence was also shown for an equivalent mechanism in human cancer cells; The et al., Nat Commun, 2014, not cited).

Reviewer #3 (Remarks to the Author):

In this technologically sophisticated study, Liu et al. analyze the sequence of G1 progression events in cells treated to acutely inhibit Cyclin D-Cdk4/6. The classical pathway requires mitogen dependent activation of the CyclinD-CDK4/6 complex. However, it has been previously observed that the genetic modifications that make the above activation step impossible do not prevent all cells from successfully entering S phase. Here, the authors ask how a subpopulation of cells with the acutely

inhibited CyclinD-CDK4/6 complex are capable of the same transition. Furthermore, they focus on elucidating the order of events in the Cyclin D-Cdk4/6-independent pathway. Not surprisingly, they observe that cells with inhibited CyclinD-CDK4/6 are proliferating due to the activation of the CyclinE-CDK2 complex as has been reported previously by others. An interesting and novel observation is that the cells choosing the alternative pathway show a reverse order of the activation of two major switches of the G1 to S transition compared to previous work with otherwise unperturbed cells. Namely, activating CyclinE-CDK2 and switching off the major G1 phase ubiquitin ligase APC/CCdh1 before hyperphosphorylating to inactivate Rb. These are important observations as they show that the temporal order of events in G1 phase is more flexible than was previously acknowledged.

This study is detailed and thorough. In my opinion, the authors present enough evidence corroborating the existence of the alternative order of the G1 to S transition events. However, my main concern is about how the authors choose to present their findings. The overall presentation is one of certainty and sharp classifications rather than a more nuanced description that accommodates prior work and a biological continuum. I urge the authors to consider that the binary categories used to describe the results may oversimplify a complex landscape of possibilities available to cells. Specifically, I would like the authors to address the points below:

Major points:

1. An arbitrary choice of threshold levels that gives rise to binary categories.

In many biological processes cells choose to follow mutually exclusive pathways and exist in well-defined, discrete states. For example, cells differentiate into different types. They can be proliferating or senescent. However, it does not follow that all biological processes can be accurately described using discrete categories.

The authors introduce 'two different paths – one with slowed and gradual CDK2 activation, and one with fast and persistent CDK2 activation'. In figure 2 that distinction is made based on a hard threshold with the value of 0.7. However, it is my impression that cells in figure 2 (both treated and untreated) show the whole continuous spectrum of behaviors.

(a) How was this threshold chosen? Is it the value that gives the highest separation between two classes? Could the authors provide or discuss an extended analysis of how the quantification changes when different thresholds are considered? By the same token, how were single time points after mitogen stimulation chosen for analysis (e.g. 32 hrs).

(b) Should a reader of this study think of S phase entry as a process governed by two parallel pathways or rather as a continuous spectrum of signaling events, with the classical and the alternative pathways being simply two extremes favored more or less in certain conditions? Can the authors really exclude the existence of cells in which S phase entry is driven simultaneously by both Rb-E2F switch and APC inactivation instead of one of these processes driving the other? If not, I strongly suggest revising the language of the manuscript and address these questions in the discussion section.

If the authors choose to keep the strong division into two parallel and mutually exclusive pathways, then there are two groups of cells whose existence should be addressed – points 2 and 3.

2. Alternative pathway in the untreated control cultured cells.

A temporal order of G1 progression with Rb hyperphosphorylation preceding APC inactivation was described by the same group in previous studies (Cappell 2016, Cappell 2018). All of the studies use MCF10A as the main cell line. In the study from 2016 it was reported that ‘a large time gap separates pRb-E2F activation from APCdh1 inactivation’.

In the current study, the authors present a comparison of intestinal crypts in control and Palbociclib treated mice (Fig. 4f). Interestingly, even in control animals many cells were classified as taking the alternative pathway (with low Rb phosphorylation and high levels of the APC degnon). The authors do not provide the analogous quantification in the cultured cells.

Are there cells in the untreated population of the in vitro cultures that choose the alternative pathway? If yes, what experimental differences allowed them to be detected this time while being omitted in the previous study? If not, could the authors comment what difference between cells in the intestinal crypts and cultured cell lines could possibly make them behave differently? Indeed, the use of the term “parallel” implies that cells in the same population could choose either pathway.

3. Cells in the Palbociclib treated in vitro cultures following the classical pathway.

In figure 2b 34% of cells were classified to highly increase CyclinE/CDK activity while in figure 4a, it's clearly stated that 22% of cells with minimally increased level of APC degnon show strong Rb phosphorylation in the presence of Palbociclib. It would suggest that a significant number of cells can actually choose the classical pathway even during Palbociclib treatment. How can the existence of these cells be explained?

In figure 4a, the deactivation of APC seems to happen in the minority of cells 32h post release in the presence of Palbociclib. To strengthen this figure and justify better the bins selected, please provide a distribution of APC degnon signal in the quiescent cells.

4. Inverted G1 signaling – what about the restriction point?

In the classical pathway, the Rb-E2F switch provides a molecular mechanism of the restriction point – a moment in the cell cycle that indicates the end of the mitogen-sensitive G1 sub-phase.

If the alternative pathway simply inverts the order of G1 events, the cells should maintain mitogen dependence after APC inactivation but before the Rb-E2F switch.

Could the authors test this hypothesis in mitogen withdrawal experiments? If the Rb-E2F switch is not underlying the restriction point in Palbociclib treated cells, is the concept of the restriction point even relevant in this population of cells? The authors should carefully explain to readers of this study who may also have read the prior work that they are not equating the mitogen-insensitive

restriction point with a commitment point that marks the onset of exogenous stress insensitivity. The irreversibility tested here could easily be confused with the mitogen-independence in work by these authors and others.

Minor points:

1. The authors should describe the ultimate fate of the cells that become EdU positive in the presence of palbociclib. Do these cells that were continuously labelled with EdU complete S phase and divide?
2. The authors should explain how they define cells that have just recently inactivated APC in Figure 5 and Extended Data 8 because it is not at all obvious for a reader to see the difference.
3. In graphs presenting percentage of cells that made a given transition vs time, it's usually reported one of several biological replicates – please add error bars to indicate variability between the experiments.
4. In figure legends the number of cells used is not defined for many assays, please add this information.
5. When t-test is used between pairs of observations, please indicate the pairs (for example, fig. 2e).
6. 2b – if 0.7 was used as a threshold, please add the line to the graphs for easier interpretation.
7. Line 248 – 'DNA replication' or 'EdU incorporation'.
8. The reference that Dbf4 is required for S phase entry is not the appropriate one.

For clarity, our responses to the points raised by the reviewers are inserted in blue. In cases where reviewers referenced specific panels that now have outdated labels, we have provided the new labels next to the original. All citations listed in this response can be found at the end of the document.

Reviewer #1 (Remarks to the Author):

The authors have carried out a sophisticated analysis of the regulatory system that drives entry into the mammalian cell cycle. The work clarifies the role of specific components and reveals how the system adapts in response to removal of certain key components. These results will be of interest to the many readers studying the control of cell proliferation in normal and tumor cells.

The current view of the cell cycle entry system is roughly the following: Mitogens -> Myc -> cyclin D-Cdk4/6 -> Rb phosphorylation -> E2F activation -> expression of cyclin E, A, EMI1 -> Cdh1 inactivation -> cyclin A stabilization -> S phase. This system includes numerous single and double feedbacks (for example, cyclin E-Cdk2 can also phosphorylate Rb, leading to positive feedback; double negative feedback is provided by mutual antagonism between Cdh1 and EMI1 or cyclin A). This is therefore a complex and robust regulatory switch that withstands inactivation of certain key molecules. For example, it is known that cyclin D-Cdk4/6 knockouts are not lethal, and cyclin E-Cdk2 promotes cell cycle entry under these conditions.

The current paper provides interesting and important new molecular details about how the system responds to the removal of cyclin D-Cdk4/6 activity, and how cyclin E-Cdk2 provides compensation. The authors use acute chemical inhibition of Cdk4/6 activity to explore the mechanisms by which the cell adapts to the loss of this kinase. This approach is superior to the previous approach of gene knockouts, allowing more detailed analysis of immediate effects. In the absence of cyclin D-Cdk4/6 activity, cyclin E gradually accumulates, presumably because it is a Myc target. As it accumulates, cyclin E-Cdk2 phosphorylates two targets that are relevant here: Rb (leading to E2F activation and further cyclin E/A production) and Cdh1 (leading to APC inactivation and stabilization of cyclin A). In a normal cell, cyclin D-Cdk4/6 is a specific Rb kinase and initiates Rb phosphorylation before cyclin E-Cdk2 activity accumulates; thus, Rb is phosphorylated well before Cdh1 is inactivated. In the absence of cyclin D-Cdk4/6, however, the authors show that the order of these phosphorylation events is reversed: Cdh1 is phosphorylated slightly before Rb. This defect in the sequence of events presumably results from the fact that cyclin E-Cdk2 is now the kinase primarily responsible for both phosphorylation events. It seems likely that Cdh1 is a better cyclin E-Cdk2 substrate and is modified first.

This reversal of the normal order of regulatory events leads to a new scheme in which Rb phosphorylation, rather than Cdh1 inactivation, is now the final step to cell cycle commitment. The work therefore suggests that both Rb phosphorylation (=E2F activation) and Cdh1 inactivation (=cyclin A stabilization) are essential for cell cycle entry, but their precise order is not critical. Previous work (including work from these authors) has argued that Rb phosphorylation represents the critical event for cell cycle commitment, but the

current paper provides a more nuanced and interesting view: that two major regulatory events together drive commitment.

We thank the referee for the positive review and for taking the time to review the manuscript.

Reviewer #2 (Remarks to the Author):

The results of this paper are over-interpreted. The title “A parallel cell cycle entry pathway with inverted G1 signalling and alternate point of no return” sounds like a dramatic discovery, but amounts to hype as it is not strongly supported by the data. The genetic deletion of CDK4 and/or CDK6 in various metazoan systems has shown long ago that these kinases are not required for entry into the cell cycle, although they enhance it. So the discovery that CDK4/6 inhibition slows but does not prevent entry into the cell cycle is not novel. I failed to grasp the reasoning behind the need to invoke the existence of two alternate “pathways” rather than simply a single mechanism which is slower when CDK4/6 activity is inhibited (ie redundant CDK activity, resulting in both Rb inactivation and CDH1 inactivation, at slightly different kinetics depending on whether CDK1/2/4/6 are active or only CDK1/2). This might sound less exciting but to me reflects equally well the overall conclusions from the experiments, which are complex.

We thank the reviewer for the feedback and have changed the manuscript title to “Altered G1 signaling order and commitment point in cells proliferating without CDK4/6 activity”. Regarding the relevance of the finding, we would like to argue that mechanistic details of how cells enter the cell cycle without CDK4/6 activity is not only important therapeutically, but also important for a basic understanding of how cells start the cell cycle. We agree that cells proliferating without CDK4/6 activity have been identified and studied in the past, but besides dependency on cyclin E-CDK2, the mechanism of how they inactivate Rb and APC/C remained elusive, and whether these cells can be found in normal mice without genetic loss of cyclins or CDKs was also unknown. We disagree with the reviewer’s assertion that the timing difference between Rb and APC/C^{CDH1} inactivation is slight, as it is a >3hr difference (either Rb inactivated ~3 hours before APC/C^{CDH1} or APC/C^{CDH1} inactivated ~4 hours before Rb), a long period of time in the normally fairly short G1 phase of the cell cycle. Our results further demonstrate signaling order plasticity in G1, which to our knowledge has not been reported before. Finally, we added new data demonstrating that in this alternative way of starting the cell cycle, accelerating APC/C^{CDH1} inactivation via Cdh1 knockdown also accelerates Rb inactivation (Extended Data Fig. 7a-b), suggesting that each of the two critical G1 inactivation events can reinforce the inactivation of the other.

Indeed, the single cell approach, while elegant, requires some quite complex gating, is not always easy to follow, and appears over-interpreted in places. The single cell reporters have their limitations: for example, the CDK2 activity reporter represents the ratio of the sum of

CDK1/2 activity with respect to the phosphatase activity, rather than the claimed cyclin E-CDK2 activity.

The reviewer makes a good point that cyclin E/A-CDK activity reporter represents the balance between cyclin E-CDK2 activity and phosphatase activities in G0/G1. To address this point, we performed a control experiment where we treated cells undergoing both entry mechanisms with a saturating dose of CDK1/2 inhibitor. Since we find no significant difference in the kinetics by which the reporter signals drop in the two cases (reflecting dephosphorylation of the reporter), differences in phosphatase activities cannot explain the different reporter activity we see between the two conditions (Extended Data Fig. 2e). The observed difference in the sensor signal thus reflects the difference in cyclin E-CDK2 activity during G1.

In some instances a population approach would allow to reinforce some conclusions. Below I present my analysis of the experiments.

1. In the experiments, starting with that in figure 1, the concentration of palbociclib used is 1 μ M. In these conditions, 10% of cells complete S-phase, whereas 30% progress through G1 (as assessed by APC/C inactivation) but with a strong delay compared to controls. What happens with lower or higher doses (100nM, 10 μ M)? The concentration used should be justified and evidence presented to support the claim that no residual CDK4/6 activity exists. This is critical for the interpretation of essentially all the experiments in the paper. Showing that 3 μ M rather than 1 μ M causes no further inhibition of S-phase entry is not appropriate: it says nothing about cycD-CDK4/6 activity, since S-phase entry is the consequence of normal CDK4/6 activity and is what is being measured. It would be thus be a circular argument to claim that this proves that all CDK4/6 activity is inhibited. The authors themselves suggest, for example, that cells can go into S-phase with a low or no CDK4/6 activity. A direct readout for CDK4/6 activity would be essential; for example, a blot analysis of monophosphorylated Rb species.

In response to the point on inhibitor concentration, we have added in the text that the Palbociclib IC₅₀ for CDK4 and CDK6 are <20nM¹. In addition, we have added text clarifying that if there is residual CDK4/6 activity, and the residual CDK4/6 activity is driving S phase, there should be a reduction in S phase entry when a higher concentration of CDK4/6 inhibitor was used, which we did not observe. 3 μ M of Palbociclib is the highest level that can be reversibly used in these cells since cells treated at 10 μ M have a reduced ability to proliferate after drug washout. Furthermore, the overall reduction in S phase cells during CDK4/6i treatment is also consistent with previous cyclin D knockout data in the literature. As for using bulk-cell analysis to demonstrate CDK4/6 inhibition, given that cells without CDK4/6 activity are not anymore synchronous, bulk cell assays to validate CDK4/6 activity suppression during the transient G1 period would not be conclusive. We would like to note that single-cell assays can be used to address this point. Our recent development of a CDK4/6 activity reporter shows inhibition of the reporter signal by this concentration of Palbociclib during G1². Furthermore, in figures 3c-d, we demonstrate that in cyclin E-CDK2 initiated cell-cycle entry, CDK2 activity is upregulated independent of

Rb hyperphosphorylation. Given our recent finding that CDK4/6 activity is continuously required for Rb hyperphosphorylation when CDK2 activity is $<0.9^3$, the data in figures 3c-d demonstrate that with this dose of Palbociclib, we are blocking CDK4/6 activity since there is no Rb hyperphosphorylation. And in figures 3e-f, we show that the eventual Rb hyperphosphorylation that occurs much later but before S phase is driven by a much higher level of measured CDK2 activity.

2. From the experiment in figure 1, the authors conclude that the results reveal an alternative pathway. There is no evidence for this.

We have changed the language and more specifically state that acute CDK4/6 inhibition reveals a delayed and less efficient cell-cycle entry mechanism.

3. Since, in Fig 2b, c, the total percentage of cells does not add up to 100, presumably the grey traces are the rest (this is not mentioned in the legend). Thus, even without CDK4/6i, it appears that 30% of cells do not activate CDKs. With CDK4/6i, this percentage goes up to 70%. So in both cases many cells do not use this “alternate pathway” – or, perhaps, these cells fail to reach a threshold overall activity required for S-phase entry.

We have clarified the grey traces in our legend, and the reviewer is correct that many cells do not activate CDK2 activity in this time frame. However, with enough time (70 hours), many cells eventually will enter the cell cycle, suggesting a delayed and less effective cell-cycle entry mechanism (Figure 1e).

4. I’m not convinced that the fluctuations measured by imaging genuinely reflect fluctuating CDK activity rather than some technical noise. This cannot simply be assumed without showing some other evidence for it.

This is a relevant point. We expressed in the same cell the WT CDK activity sensor as well as an alanine mutant of the CDK activity sensor, where the reporter cannot be phosphorylated and thus stays nuclear (Extended Data Fig. 2c). The experiment demonstrated that the fluctuations are not due to technical noise, and we have emphasized this more clearly in the text. In addition, we have also in the past expressed in the same cell a fluorescent protein as well as the WT CDK activity sensor and arrived at the same conclusion⁴. Finally, the fluctuations were not purely background noise since knocking down cyclin E and p21 changes the percentages of fluctuating cells significantly (Fig. 2c).

5. In fig 2c, the siRNA control cells have 14.5 high and 20.9% fluctuating CDK reporter activity, yet in fig 2b the corresponding numbers are 34.1% and 13.1%, respectively. Is this just biological variability, or does the siRNA control affect the cell cycle on its own? Either way, what can we say about this variability?

This is again a good point. Both are likely contributing. Variations in seeding density, endogenous culturing stress, serum aliquot, etc. can all affect these percentages when experiments are repeated. We also note that the transfection protocol can also introduce

stress and affect the percentages, which also accounts for the percentage drop in the si Ctrl condition (we have included this in the text).

6. There are no WB controls shown for the efficiency of siRNA-mediated knockdown in any experiment.

Thank you for pointing this out. We have added western blot controls.

7. In fig 2e, it is not explained how c-Myc levels were quantified and the original data is not shown, so how the authors conclude that a 50% increase in c-Myc causes the observed effects is unclear.

We again thank the reviewer for pointing this out. c-Myc was quantified via protein immunofluorescence and we have now added this in the text.

8. I don't understand the conclusion that "c-Myc and cyclin D-CDK4/6 activities are driving forces that can promote cell-cycle entry through two different paths – one with slowed and gradual CDK2 activation, and one with fast and persistent CDK2 activation". Why should the different kinetics of CDK2 activation when CDK4/6 or Myc are manipulated necessarily reflect different pathways, rather than simply a less active pathway? Besides, c-Myc is a target gene controlled by cyclin D-CDK4/6 activity, so imagining what these pathways would look like is difficult.

In our system, c-Myc protein level is unlikely to be controlled by CDK4/6 activity (see Myc protein levels in Figure 2e, tested with no TMP condition). To address this point, we have now also included a more detailed protein time course experiment where we quantified c-Myc in cells with 2n DNA coming out of mitogen release with and without CDK4/6 inhibitor (Extended Data Fig. 3a).

We agree that different readers may have different views on what should be termed a pathway. We have thus in the revised manuscript avoided using the term pathway and mostly used terms such as cyclin D-CDK4/6-initiated and cyclin E-CDK2-initiated cell-cycle entry. We do note that without CDK4/6 activity, cells start the cell cycle with extra dependence on c-Myc, and proceed through G1 with a different order of events. These two different ways to initiate cell-cycle entry can be seen in wild-type crypt cells, and our new data in the revised manuscript suggests that without CDK4/6 activity, APC/C^{CDH1} inactivation helps lead to Rb inactivation as opposed to the other way around where Rb inactivation leads to APC/C^{CDH1} inactivation when CDK4/6 is present.

9. During the description of the experiment shown in figure 3, the Narasimha paper (eLife 2014, cited ref 37) is presented in conjunction with data on RB S807/11 phosphorylation, to support the argument that with CDK4/6 inhibition, cyclin E-Cdk2 is upregulated by c-Myc. I am not convinced by this experiment, for several reasons. First, the data tend to suggest that CDK4/6 can monophosphorylate Rb on S807/811, as expected from the Narasimha paper. Thus, S807/11ph may not distinguish between Rb mono and hyperphosphorylation. Second, no data on cyclin E-Cdk2 activity is presented, the only

data shown are single cell traces of DHB cytoplasmic/nuclear ratio. This perhaps corresponds to Cyclin E-CDK2 activity in normal circumstances (or rather, the ratio of CDK1/2 activity to protein phosphatase activity), but this may not be the case when CDK4/6 is inhibited. Rather than doing a single cell analysis, a population analysis of CDK2-cyclin E activity would resolve this issue. The advantage of the single cell approach is not obvious here.

Our recent publication demonstrated the phospho-Rb S807/811 used here is a marker for Rb hyperphosphorylation and that this phosphorylation and E2F target activation can be mediated in G1 by CDK4/6 activity (shown for example in MEFs lacking all four cyclin A and E genes)³. The use of the antibody as a marker for Rb hyperphosphorylation and inactivation was also confirmed by single-cell co-staining with other phospho-Rb markers and also functionally via single-cell mRNA FISH and Rb chromatin binding measurements³. This recent publication also showed that the same phospho-Rb S807/S811 antibody could not distinguish the small shift in an immunostaining analysis between unphosphorylated Rb and the small fraction of mono-phosphorylated Rb (both forms of Rb are able to inhibit E2F). We have included this reference in the text and believe that this clarifies the point of the reviewer. Regardless, most if not all sites need to be phosphorylated for Rb to be inactivated (as is the case in S/G2 cells in Fig. 3c), and absence of two key phosphorylation sites demonstrate that cyclin E-CDK2 activity is upregulated without Rb hyperphosphorylation in CDK4/6 inhibitor treated cells.

The reviewer brings up a good point about the balance between kinase and phosphatase activity. In addition to the DHB reporter being sensitive to cyclin E knockdown in the presence of CDK4/6i (Figure 2c), we have added data where we treated cells proliferating with or without CDK4/6i a saturating dose of CDK1/2 inhibitor (Extended Data Fig. 2e). The sensor decreased with similar fast kinetics in the two conditions, suggesting that the phosphatase activities acting on the sensor are comparable for the two conditions.

10. The experiment described in figure 4 is very hard to understand. First, if cells are analysed at 32h after CDK4/6 inhibition, yet only 20h after control (DMSO) treatment, this surely means that APC/C inactivation is much later than without CDK4/6 inhibition? So the starting point is different. Second, it cannot be taken for granted, without generating a circular argument, that the level of degron expression can be converted into time after APC/C inactivation both in control and CDK4/6i conditions, because the reporter might be transcribed/translated at different rates in the latter. Even assuming all is OK, it could simply be that APC/C is inactivated at lower CDK 1/2 levels than are required to hyperphosphorylate Rb, whereas everything occurs more or less simultaneously when CDK4 and 6 are active. I don't understand the conclusion of an "inversion" of these events, much less "inverted G1 signalling" or what the real significance of this observation is.

We thank the reviewer for pointing this out and have changed the text to improve clarity. The reviewer is correct that without CDK4/6 activity, APC/C^{CDH1} inactivates at a much later time compared to cells with CDK4/6 activity. The different time points were chosen to get sufficient number of cells that just started to express APC/C degron in both cases.

Notably, despite this delayed APC/C^{CDH1} inactivation, cells proliferating without CDK4/6 activity still have a much earlier APC/C^{CDH1} inactivation relative to Rb inactivation in cells treated with CDK4/6i.

We disagree with the “more or less the same” argument since APC/C^{CDH1} inactivates ~3 hours after Rb inactivation when CDK4/6 activity is present (Figure 6b), which is a long time in terms of G1. We also believe that prior to this manuscript, observing that there is now a ~ 4 hour APC/C inactivation prior to Rb inactivation in CDK4/6 inhibited cells has not been reported, and we also report this temporal plasticity in a wild type, *in vivo* context. To further reinforce this point, we have included the new siCdh1 data mentioned before to demonstrate that in the cyclin E-CDK2 initiated pathway, the sequence of events is different, and the causal link is also flipped between Rb and APC/C inactivation (Extended Data Fig. 7a-b). Finally, this change of sequence showed that the stress commitment point is not always at the time when APC/C^{CDH1} is first inactivated.

11. It gets even more confusing in Fig 4b, where E2F1 transcripts are analysed, since it is stated that the way the experiment was done was to gate cells with high CDK1/2 activity but low APC/C degnon. Yet in Extended data Fig 5a it is shown that APC/C activity is switched off at similar CDK1/2 activity levels whether CDK4/6 are active or inhibited. So in high CDK1/2 activity, APC/C degnon levels should be high.

Thank you for pointing this out. We have changed the text to improve clarity. CDK2 is activated first before APC/C is turned off in both mechanisms of cell-cycle entry, the data was thus gated for the time period in between CDK2 activation and APC/C^{CDH1} inactivation.

12. The *in vivo* analysis in fig 4f (now Figure 5) is not very convincing, as the interpretation is based on a small proportion of cells that are very hard to see (nuclei are not marked). It may also be an issue that after hyperphosphorylation, Rb protein itself may be decreased, and in these IF experiments there is no indication of the ratio of phospho Rb to total Rb.

Thank you for pointing this out. We apologize for the small panels and have increased the size, marked key nuclei, and included zoomed in images. We have tried using multiple ways to measure total Rb content in these *in vivo* slices but could not find total Rb antibodies that showed a significant signal:

phospho-Rb (S807/S811)

Rb

We have included in the revision a time course plot of Rb vs phospho-Rb in MCF-10A cells, and showed that Rb level does not decrease after phosphorylation (Extended Data Fig. 8c).

13. Fig 5f-h (now Figure 6f-h) is even harder to understand. The explanation is telegraphic, to say the least, about what was actually done, how the data tell us anything about Rb inactivation (which was not assessed as far as I can tell), and how commitment was defined. It looks suspiciously like a circular argument: commitment being the point beyond which APC/C is no longer reactivated after application of salt. Yet that is what is being measured to conclude this.

Thank you for pointing this out. We have now added more details in the text and simplified the panels to improve clarity (in addition to adding log scaled plots). Rb was not assessed directly because we do not have a live-cell reporter of Rb inactivation. In short, previous work identified irreversible APC/C^{CDH1} inactivation to be the point of no return/point of stress insensitivity, which allows proteins such as cyclin A to permanently build up without interruption, thus ensuring a one-directional S phase and safeguarding genome integrity^{5,6}. In cells with CDK4/6 activity, irreversible APC/C^{CDH1} inactivation occurs early at the onset of APC/C^{CDH1} inactivation. However, in cells proliferating without CDK4/6 activity, APC/C^{CDH1} is reversible until the time of Rb inactivation, which occurs later, thus the point of no return is also not crossed until then.

Other points:

1. In the introduction, the current model for cell cycle entry is carefully presented, justifying the study; but this is nevertheless somewhat misleading. It is said that cyclin D-CDK4/6 complexes lead to Rb inactivation. Indeed, it was previously thought that CycD-CDK4/6 progressively phosphorylate and inactivate Rb, thus de-repressing cyclin E transcription, triggering a positive feedback loop since cycE-CDK2 complexes complete inactivation of Rb. However, more recent work by the Dowdy lab showed, in non-transformed cells, that this model is incorrect; instead, cycD-CDK4/6 complexes monophosphorylate Rb, and this monophosphorylated species is active as a transcriptional

repressor; Rb is then quantitatively hyper-phosphorylated, and inactivated, by CycE complexes later. This model should be presented in the introduction. It is rather presented later in the results section (see above, discussion of experiment shown in figure 3) but how the current results fit with this model is not discussed.

Recent work from our lab and others presented evidence suggesting that Rb can be inactivated by cyclin D-CDK4/6^{3,7}. For example, MEFs lacking all four cyclin E and A genes still hyperphosphorylate Rb and induced E2F targets in a CDK4/6 activity dependent manner without showing any measurable CDK2 activity³. Nevertheless, we agree that more work is needed to sort out the question of how Rb is inactivated in different contexts. To our knowledge, most groups agree that cyclin D-CDK4/6 can phosphorylate Rb, and that cyclin D-CDK4/6 activation contributes to the eventual inactivation of Rb. We avoided an extensive discussion of how Rb is inactivated because we did not think it was critical for the introduction of this manuscript, nevertheless, the reviewer brings up a good point and we have now focused more on where the models agree, and included the citations of these different proposed mechanisms.

2. The conclusion of the inversion of the order of CDH1 inactivation and Rb inactivation in the absence of CDK4/6 activity possibly reflects specificity of CDK4/6 complexes for CDH1 itself, as shown for homologues in nematodes by the Van den Heuvel lab (evidence was also shown for an equivalent mechanism in human cancer cells; The et al., Nat Commun, 2014, not cited).

We have cited this work.

Reviewer #3 (Remarks to the Author):

In this technologically sophisticated study, Liu et al. analyze the sequence of G1 progression events in cells treated to acutely inhibit Cyclin D-Cdk4/6. The classical pathway requires mitogen dependent activation of the CyclinD-CDK4/6 complex. However, it has been previously observed that the genetic modifications that make the above activation step impossible do not prevent all cells from successfully entering S phase. Here, the authors ask how a subpopulation of cells with the acutely inhibited CyclinD-CDK4/6 complex are capable of the same transition. Furthermore, they focus on elucidating the order of events in the Cyclin D-Cdk4/6-independent pathway. Not surprisingly, they observe that cells with inhibited CyclinD-CDK4/6 are proliferating due to the activation of the CyclinE-CDK2 complex as has been reported previously by others. An interesting and novel observation is that the cells choosing the alternative pathway show a reverse order of the activation of two major switches of the G1 to S transition compared to previous work with otherwise unperturbed cells. Namely, activating CyclinE-CDK2 and switching off the major G1 phase ubiquitin ligase APC/CCdh1 before hyperphosphorylating to inactivate Rb. These are important observations as they show that the temporal order of events in G1 phase is more flexible than was previously acknowledged.

This study is detailed and thorough. In my opinion, the authors present enough evidence corroborating the existence of the alternative order of the G1 to S transition events. However, my main concern is about how the authors choose to present their findings. The overall presentation is one of certainty and sharp classifications rather than a more nuanced description that accommodates prior work and a biological continuum. I urge the authors to consider that the binary categories used to describe the results may oversimplify a complex landscape of possibilities available to cells. Specifically, I would like the authors to address the points below:

Major points:

1. An arbitrary choice of threshold levels that gives rise to binary categories.

In many biological processes cells choose to follow mutually exclusive pathways and exist in well-defined, discrete states. For example, cells differentiate into different types. They can be proliferating or senescent. However, it does not follow that all biological processes can be accurately described using discrete categories.

The authors introduce ‘two different paths – one with slowed and gradual CDK2 activation, and one with fast and persistent CDK2 activation’. In figure 2 that distinction is made based on a hard threshold with the value of 0.7. However, it is my impression that cells in figure 2 (both treated and untreated) show the whole continuous spectrum of behaviors. (a) How was this threshold chosen? Is it the value that gives the highest separation between two classes? Could the authors provide or discuss an extended analysis of how the quantification changes when different thresholds are considered? By the same token, how were single time points after mitogen stimulation chosen for analysis (e.g. 32 hrs).

We thank the reviewer for the helpful feedback. The 0.7 was chosen based on the negative control ($t=0$) where no mitogens were added (we have now added this in the text to improve clarity), and also with the knowledge that most cells entered S phase at around 0.9^{3,5,8}. We thought selecting a specific threshold offers a simple way to quantitatively communicate the difference between the two conditions. To address this point, with the same dataset, we changed the threshold to 0.6 and 0.8 and showed that the conclusion remains the same. We have included the data in Extended Data Figure 2b.

We would like to point out that the blue cells (cyclin E/A-CDK activity high) in the two conditions in figure 2b are not exactly the same. Even though they eventually reach >0.7 after 24hrs, CDK4/6i treated cells have delayed activation and the slope is also slower. Both blue and green cells in the CDK4/6i condition exhibit slowed and non-persistent behavior, but the blue cells eventually reach high enough of CDK activity to continue the cell cycle in the time period we monitored. We have made the slower kinetics clearer in the text. The 24 and 32hr time points for the CDK4/6 inhibited condition (such as figure 3c and figure 4a) are chosen based on analysis like the one in figure 6b to maximize number of cells we can analyze.

- (b) Should a reader of this study think of S phase entry as a process governed by two

parallel pathways or rather as a continuous spectrum of signaling events, with the classical and the alternative pathways being simply two extremes favored more or less in certain conditions? Can the authors really exclude the existence of cells in which S phase entry is driven simultaneously by both Rb-E2F switch and APC inactivation instead of one of these processes driving the other? If not, I strongly suggest revising the language of the manuscript and address these questions in the discussion section.

We cannot rule out that the CDK4/6-initiated and CDK2-initiated conditions we compared represent two extreme cases of a continuum in different cell types/conditions. We have thus revised our language (for example, removing usage of the word “parallel”) and included such possibility in the discussion. In the revised manuscript, we also included new data where we accelerated APC/C^{CDH1} inactivation in cells without CDK4/6 activity, and observed an increase in cells with Rb inactivation (Extended Data Fig. 7a-b). This demonstrates that without CDK4/6 activity, APC/C^{CDH1} inactivation helps Rb inactivate as opposed to the other way around in cells that do have CDK4/6 activity. This data may suggest that in most cases one inactivation is triggered first due to a relative activity difference and then helps the inactivation of the other, which makes a simultaneous inactivation model of both Rb and APC/C^{CDH1} potentially less likely.

If the authors choose to keep the strong division into two parallel and mutually exclusive pathways, then there are two groups of cells which existence should be addressed – points 2 and 3.

2. Alternative pathway in the untreated control cultured cells.

A temporal order of G1 progression with Rb hyperphosphorylation preceding APC inactivation was described by the same group in previous studies (Cappell 2016, Cappell 2018). All of the studies use MCF10A as the main cell line. In the study from 2016 it was reported that ‘a large time gap separates pRb-E2F activation from APCCdh1 inactivation’.

In the current study, the authors present a comparison of intestinal crypts in control and Palbociclib treated mice (Fig. 4f) (now Fig. 5c). Interestingly, even in control animals many cells were classified as taking the alternative pathway (with low Rb phosphorylation and high levels of the APC degra). The authors do not provide the analogous quantification in the cultured cells.

Are there cells in the untreated population of the in vitro cultures that choose the alternative pathway? If yes, what experimental differences allowed them to be detected this time while being omitted in the previous study? If not, could the authors comment what difference between cells in the intestinal crypts and cultured cell lines could possibly make them behave differently? Indeed, the use of the term “parallel” implies that cells in the same population could choose either pathway.

In unperturbed cultured cells, we still observe some rare cells that show signatures of this alternate path (DMSO condition, figures 3c-d, where CDK2 activity is initiated yet some cells still have low phospho-Rb signal; we have included this in our discussion). The reason

it was not detected at the time of Cappell 2016 was because the cells were plotted like Figure 6b of this manuscript (as bulk percentages that can mask small populations). Below is a panel from Cappell et al., 2016 for comparison⁵:

The percentage of cells undertaking the alternate sequence in unperturbed cultured cells is low compared to wild type intestinal crypts, and this could be due to growth factor concentrations, long term selection in culture, or other reasons. It is also possible that different cell types may have different percentages. In crypts, for example, it is well-known that cells rely heavily on Wnt signaling, which upregulates c-Myc⁹.

3. Cells in the Palbociclib treated in vitro cultures following the classical pathway.

In figure 2b 34% of cells were classified to highly increase CyclinE/CDK activity while in figure 4a, it's clearly stated that 22% of cells with minimally increased level of APC degnon show strong Rb phosphorylation in the presence of Palbociclib. It would suggest that a significant number of cells can actually choose the classical pathway even during Palbociclib treatment. How can the existence of these cells be explained?

Cells without CDK4/6 activity eventually build up high cyclin E/A-CDK activity, though at a slower rate of increase (with occasional fluctuations), thus explaining most of the CDK high cells in figure 2b. In figure 4a and Extended Data figure 5b (cells out of mitogen release and asynchronously cycling cells), the presence of some recently-inactivated APC/C cells with high phospho-Rb signal is at least partially due to the imperfection of calling APC/C-degnon buildup (we note that due to fluorescent protein maturation time, characterized in Cappell et al. 2016, we cannot detect APC/C degnon until ~48 minutes after it starts building up). Based on this data alone, however, we do not know for certain whether these cells inactivated Rb prior to APC/C^{CDH1}, but we can conclude that the majority of cells inactivated APC/C^{CDH1} prior to Rb.

In figure 4a, the deactivation of APC seems to happen in the minority of cells 32h post release in the presence of Palbociclib. To strengthen this figure and justify better the bins selected, please provide a distribution of APC degnon signal in the quiescent cells.

The plot below shows the APC/C degnon histogram analysis in the quiescent cells. We would like to note that in non-live cell experiments, the unreleased control has been starved for an extra 32hrs (for a total of ~80hrs). This longer starvation can be avoided when cells

are imaged and followed live, since we can use $t=0$ as a control to determine that APC/C^{CDH1} is off. In the histogram analysis, we did not do a live-cell experiment to get more cells for the binning analysis. Using this analysis, the additional starvation time reduces basal metabolic activity so that the residual sensors' "background" (driven by the constitutive promoter) is lower. Thus, these extra-starved cells show slightly lower basal APC/C degron levels than cells just a few hours after mitogen release, where we know from the literature that APC/C^{CDH1} is still active. This difference is very small (<5 RFUs in an assay where the dynamic range maxes out at >1000). However, since the histograms are shown in log₂ scale, the little difference in RFU appears pronounced. We thus estimate the APC/C^{CDH1} off state based on the two histogram peaks. We note that this type of peak-based estimation tends to estimate a slightly later time of APC/C^{CDH1} inactivation, but despite this later estimation, APC/C^{CDH1} inactivation still occurs prior to Rb inactivation. We did not include this longer discussion in the main manuscript due to the manuscript length limit.

4. Inverted G1 signaling – what about the restriction point?

In the classical pathway, the Rb-E2F switch provides a molecular mechanism of the restriction point – a moment in the cell cycle that indicates the end of the mitogen-sensitive G1 sub-phase.

If the alternative pathway simply inverts the order of G1 events, the cells should maintain mitogen dependence after APC inactivation but before the Rb-E2F switch.

Could the authors test this hypothesis in mitogen withdrawal experiments? If the Rb-E2F switch is not underlying the restriction point in Palbociclib treated cells, is the concept of the restriction point even relevant in this population of cells? The authors should carefully explain to readers of this study who may also have read the prior work that they are not equating the mitogen-insensitive restriction point with a commitment point that marks the onset of exogenous stress insensitivity. The irreversibility tested here could easily be confused with the mitogen-independence in work by these authors and others.

We have followed the reviewer's recommendation and clarified that we are not equating mitogen-insensitivity with stress-insensitivity. In addition, with our own and other recently published work, we now believe that cell-cycle entry is best described by a time-dependent probabilistic model of restriction point, as opposed to a sharp time point^{3,8,10,11}. We have

also tested in this revision mitogen withdrawal in cells proliferating without CDK4/6 activity, and found that as expected, these cells remain sensitive to mitogen after APC/C^{CDH1} inactivation (Extended Data Fig. 10).

Minor points:

1. The authors should describe the ultimate fate of the cells that become EdU positive in the presence of palbociclib. Do these cells that were continuously labelled with EdU complete S phase and divide?

We did not track the cells in the continuous EdU labelling experiment in Figure 1c, so we do not know the exact percentage of cells that went through mitosis. We do know that most cells do not finish mitosis by this time point (can be inferred from Figure 1b, where in the DMSO condition, the drop in APC/C degen is not evident until ~24hrs). We also believe that the percent of cells undergoing mitosis will be lower than usual in this condition, since continuous EdU labelling is ultimately stressful to cells. We do note that CDK4/6 inhibited cells eventually complete mitosis, and we have included this in the text.

2. The authors should explain how they define cells that have just recently inactivated APC in Figure 5 (now Figure 6) and Extended Data 8 (now Extended Data 9) because it is not at all obvious for a reader to see the difference.

We apologize for this lack of clarity, and have included log₂ plots to help the readers assess the inactivation in addition to text clarification.

3. In graphs presenting percentage of cells that made a given transition vs time, it's usually reported one of several biological replicates – please add error bars to indicate variability between the experiments.

We have included error bars for the percent cells vs time plots.

4. In figure legends the number of cells used is not defined for many assays, please add this information.

We have included this information, but will note that in figures with a multitude of conditions (such as histograms in Figure 4a, bar graphs in Figure 1g, or traces in Figure 6d), we simply listed the minimum number of cells in all of the conditions for brevity reasons. If the reviewer would still like us to list the exact number for these conditions, we can still do so.

5. When t-test is used between pairs of observations, please indicate the pairs (for example, fig. 2e).

6. 2b – if 0.7 was used as a threshold, please add the line to the graphs for easier interpretation.

7. Line 248 – ‘DNA replication’ or ‘EdU incorporation’.

8. The reference that Dbf4 is required for S phase entry is not the appropriate one.

We have adjusted the text and figures accordingly for points 5 to 8 and thank the reviewer for their time.

1. Fry, D. W. *et al.* Specific inhibition of cyclin-dependent kinase 4/6 by PD 0332991 and associated antitumor activity in human tumor xenografts. *Mol Cancer Ther* **3**, (2004).
2. Yang, H. W. *et al.* Stress-mediated exit to quiescence restricted by increasing persistence in CDK4/6 activation. *Elife* **9**, (2020).
3. Chung, M. *et al.* Transient Hysteresis in CDK4/6 Activity Underlies Passage of the Restriction Point in G1. *Mol. Cell* **76**, 562-573.e4 (2019).
4. Daigh, L. H., Liu, C., Chung, M., Cimprich, K. A. & Meyer, T. Stochastic Endogenous Replication Stress Causes ATR-Triggered Fluctuations in CDK2 Activity that Dynamically Adjust Global DNA Synthesis Rates. *Cell Syst.* (2018). doi:10.1016/j.cels.2018.05.011
5. Cappell, S. D., Chung, M., Jaimovich, A., Spencer, S. L. & Meyer, T. Irreversible APCCdh1 Inactivation Underlies the Point of No Return for Cell-Cycle Entry. *Cell* **166**, 167–180 (2016).
6. Barr, A. R., Heldt, F. S., Zhang, T., Bakal, C. & Novák, B. A Dynamical Framework for the All-or-None G1/S Transition. *Cell Syst.* **2**, 27–37 (2016).
7. Topacio, B. R. *et al.* Cyclin D-Cdk4,6 Drives Cell-Cycle Progression via the Retinoblastoma Protein's C-Terminal Helix. *Mol. Cell* **74**, 758-770.e4 (2019).
8. Spencer, S. L. *et al.* The Proliferation-Quiescence Decision Is Controlled by a Bifurcation in CDK2 Activity at Mitotic Exit. *Cell* **155**, 369–383 (2013).
9. Myant, K. & Sansom, O. J. Wnt/Myc interactions in intestinal cancer: Partners in crime. *Experimental Cell Research* **317**, 2725–2731 (2011).
10. Yang, H. W., Chung, M., Kudo, T. & Meyer, T. Competing memories of mitogen and p53 signalling control cell-cycle entry. *Nature* **549**, 404–408 (2017).
11. Min, M., Rong, Y., Tian, C. & Spencer, S. Temporal integration of mitogen history in mother cells controls proliferation of daughter cells. *Science (80-.).* **368**, eaay8241 (2020).

REVIEWERS' COMMENTS

Reviewer #1 (Remarks to the Author):

As in my previous review 14 months ago, I remain positive about the value of this paper in providing additional depth to our understanding of the important mechanisms that commit the cell to division.

Reviewer #2 (Remarks to the Author):

The revised version of the paper by Liu et al. has toned down the language about “alternative pathways”, for which I do not believe they show strong evidence (this addresses one of my major concerns). However, my second major criticism has not been addressed at all: namely, the conclusion that there is no residual CDK4/6 activity in cells treated with 1 μ M palbociclib, for which they show no evidence other than a circular argument (they increase the concentration to 3 μ M and do not see differences in their reporter – which they are claiming is independent of CDK4/6 activity). This is a critical issue, as it is quite possible that there IS residual CDK4/6 activity at 1 μ M. An argument offered, that this is far higher than the IC50 of 20nM, is not relevant: the IC50 is the concentration in vitro at which 50% of activity remains (which is any way uninformative without giving the ATP concentration in the assay - typically 20 μ M, i.e. far less than the mM ATP concentration within cells). Experimental evidence shows that within cells treated with 1 μ M (and 3 μ M) there is very likely residual CDK4, and almost certainly CDK6, activity, as shown by the CETSA assays of Martinez Molina et al., Science, 2013 (see reproduction of this data in the attached document), where it takes around 20 μ M to fully inhibit both kinases.

Thus, while I do agree that this paper does show the interesting result that the precise order of APC/Cdh1 inactivation and Rb inactivation is not critical for cell cycle commitment, and the kinetics of their respective inactivation are altered when CDK4/6 activity is reduced, I still feel that in order to be published it is essential to address this issue. Either similar assays with different concentrations of palbociclib from 0.1-10 μ M could be done (I don't buy the argument that 10 μ M cannot be used since it irreversibly inhibits cell cycle entry, since in the assays done with 1 μ M, and even to some extent the controls, many cells also are irreversibly arrested) or CETSA assays could be done. In my view it would not preclude publication even if such experiments did provide evidence for residual CDK4/6 activity, but it would make the interpretations more reliable. Since these experiments would not be time consuming or technically difficult, I cannot find any justification for not doing such experiments other than the fear that they may not support the authors' arguments. At the very least, it must be carefully presented that we are looking at cells in which CDK4/6 activity is reduced, but not necessarily eliminated.

My other concerns have largely been addressed, but for me this still cannot be published in its current state without potentially misleading readers.

Reviewer #3 (Remarks to the Author):

I am generally satisfied with this revision and congratulate the authors on an interesting study. I do have one suggestion (not demand) for the final version however partly in concordance with the response to another reviewer: I suggest including a sentence that explicitly explains to the reader how this staining protocol is not sensitive to Rb monophosphorylation, so the quantification is inferred to be primarily hyperphosphorylated Rb. The new text make the claim but does not address the monophosphorylation specifically which may be on the mind of many readers.

The authors' response is marked in blue.

Reviewer #1 (Remarks to the Author):

As in my previous review 14 months ago, I remain positive about the value of this paper in providing additional depth to our understanding of the important mechanisms that commit the cell to division.

We thank the reviewer for their time.

Reviewer #2 (Remarks to the Author):

The revised version of the paper by Liu et al. has toned down the language about “alternative pathways”, for which I do not believe they show strong evidence (this addresses one of my major concerns). However, my second major criticism has not been addressed at all: namely, the conclusion that there is no residual CDK4/6 activity in cells treated with 1 μ M palbociclib, for which they show no evidence other than a circular argument (they increase the concentration to 3 μ M and do not see differences in their reporter – which they are claiming is independent of CDK4/6 activity). This is a critical issue, as it is quite possible that there IS residual CDK4/6 activity at 1 μ M. An argument offered, that this is far higher than the IC₅₀ of 20nM, is not relevant: the IC₅₀ is the concentration in vitro at which 50% of activity remains (which is any way uninformative without giving the ATP concentration in the assay - typically 20 μ M, i.e. far less than the mM ATP concentration within cells). Experimental evidence shows that within cells treated with 1 μ M (and 3 μ M) there is very likely residual CDK4, and almost certainly CDK6, activity, as shown by the CETSA assays of Martinez Molina et al., Science, 2013 (see reproduction of this data in the attached document), where it takes around 20 μ M to fully inhibit both kinases. Thus, while I do agree that this paper does show the interesting result that the precise order of APC/Cdh1 inactivation and Rb inactivation is not critical for cell cycle commitment, and the kinetics of their respective inactivation are altered when CDK4/6 activity is reduced, I still feel that in order to be published it is essential to address this issue. Either similar assays with different concentrations of palbociclib from 0.1-10 μ M could be done (I don't buy the argument that 10 μ M cannot be used since it irreversibly inhibits cell cycle entry, since in the assays done with 1 μ M, and even to some extent the controls, many cells also are irreversibly arrested) or CETSA assays could be done. In my view it would not preclude publication even if such experiments did provide evidence for residual CDK4/6 activity, but it would make the interpretations more reliable. Since these experiments would not be time consuming or technically difficult, I cannot find any justification for not doing such experiments other than the fear that they may not support the authors' arguments. At the very least, it must be carefully presented that we are looking at cells in which CDK4/6 activity is reduced, but not necessarily eliminated.

My other concerns have largely been addressed, but for me this still cannot be published in its current state without potentially misleading readers.

1. Reviewer 2 referenced the cellular thermal shift assay (CETSA) as evidence that 20 μ M of Palbociclib (CDK4/6i in this manuscript) is required to completely block CDK4/6 activity. We think the reviewer is referencing figure 4a in Molina et al. 2013, as shown below along with its figure legend (PD0332991 is Palbociclib):

Fig. 4. Monitoring of drug and target specificity. (A) ITDR_{CETSA} at 45°C for CDKs in intact cells showing specificity for PD0332991 to CDK4 and CDK6 over CDK2 and CDK9. (B) ITDR_{CETSA} at 56°C for the

This assay is conducted at 45°C, where on/off rates of the molecule and targets are significantly affected, and will always show higher concentrations than the physiological binding temperature of 37°C. In fact, Molina et al. stated in the manuscript: “Unlike traditional dose-response experiments in which half-saturation points are related to affinities, the response here is typically reached at higher drug concentrations” and “Due to the higher temperatures of the experiment and the continuous loss of protein due to precipitation during the heating stage, this value will need further deconvolution to allow better determination of drug affinity at 37°C.” Thus, though this assay can be used to determine relative target specificity, it cannot be used to justify the 20 μ M claim.

Furthermore, this remains a bulk-cell assay where subpopulations can be masked. CETSA also measures target binding, but does not necessarily reflect the complexity of CDK4/6 biology. For example, there are likely many inactive CDK4/6 complexes (such as ones that aren’t bound to cyclin D), and these inactive CDK4/6 can also bind Palbociclib. How Palbociclib binds CDK4/6 in respect to p21 and p27 (both can form activator or repressor CDK4/6 complexes) also remains an active area of research.

The best way to measure inhibition of CDK4/6 activity in cells then is by measuring its substrates and downstream cellular functions in cells (preferably via single-cell methods). Rb(p-S807/S811) has been extensively established in our lab as a sensitive readout for CDK4/6 activity in G1 (Chung et al., 2019, citation in manuscript), and we show that our dose of 1 μ M CDK4/6i inhibits the phosphorylation of Rb in these cells (Chung et al., 2019; figure 3c-d in this manuscript). We further show that increasing the dose 3-fold has no additional effects on S-phase entry (the relevant downstream function of CDK4/6 activity, figure 1d). These data thus show that a saturating inhibitory effect of CDK4/6 towards Rb has been reached in these cells.

2. Reviewer 2 also referenced figure 1d in our manuscript, where we increased the concentration and refresh rate of CDK4/6i and saw no increased effect on inhibiting EdU incorporation/S-phase entry. Reviewer 2 stated that “...they increase the concentration to 3 μ M and do not see differences in their reporter – which they are claiming is independent of CDK4/6 activity”. Specifically, we stated in the manuscript: “We reason that if there is residual CDK4/6 activity that is driving cells into S phase, refreshing the drug more frequently or titrating up the drug

should decrease the percentage of S phase cells. However, entry into S phase was independent of the drug refreshing rate and was also not further inhibited by a three-fold increase in inhibitor concentration... arguing that there is only minimal CDK4/6 activity contribution in S-phase entry.” Our data demonstrates that S-phase entry is dependent on CDK4/6 activity up to a certain point (for example, comparing 0 to 1 μ M of CDK4/6i, we see a reduction in S-phase entry (figure 1d)); however, 3 μ M of CDK4/6i does not further prevent cell-cycle entry compared to 1 μ M, demonstrating that S-phase entry at this point is no longer dependent on an additional three-fold higher inhibition of CDK4/6 activity, and that we have saturated the effect of CDK4/6 inhibition on cell-cycle entry.

3. Like all other known kinase inhibitors, increasing the inhibitor concentration will eventually result in undesirable off-target effect which we start to observe at 10 μ M of Palbociclib. Unlike 1 μ M Palbociclib treatment, when we treat cells long-term with 10 μ M Palbociclib, we do not see Rb re-phosphorylation after washing out the inhibitor, which we interpret as an irreversible off-target damage of Palbociclib. In the case of Palbociclib and other CDK inhibitors, this likely results from the inhibition of one of the many CDKs, such as transcriptional CDKs, that can be targeted at higher levels of all known CDK inhibitors - which creates cellular stress; such kinase inhibition can block cell-cycle entry through a number of stress mechanisms. We therefore argue that titrating Palbociclib higher than 3 μ M cannot in our cells be used for making a conclusion about CDK4/6 since toxicity and off-target effects become confounding factors. Again, we would like to clarify that the only well-established target of CDK4/6, Rb, is already dephosphorylated in G1 at a dose of 1 μ M Palbociclib both by western blotting and immunostaining analysis (Chung et al., 2019; figure 3c-d in this manuscript).

In the revised manuscript, we have further clarified these points. Specifically, we emphasized the functional aspect of CDK4/6 activity and that we do not detect residual CDK4/6 activity against Rb that mediates CDK4/6-induced cell-cycle entry. We believe this should clarify the point regarding this issue. We thank the reviewer again for their time.

Reviewer #3 (Remarks to the Author):

I am generally satisfied with this revision and congratulate the authors on an interesting study. I do have one suggestion (not demand) for the final version however partly in concordance with the response to another reviewer: I suggest including a sentence that explicitly explains to the reader how this staining protocol is not sensitive to Rb monophosphorylation, so the quantification is inferred to be primarily hyperphosphorylated Rb. The new text make the claim but does not address the monophosphorylation specifically which may be on the mind of many readers.

We have included the suggestion and thank the reviewer for their time.